# PRIOR: Personalized Prior for Reactivating the Information Overlooked in Federated Learning.

**Mingjia Shi**[1*]     **Yuhao Zhou**[1]     **Kai Wang**[2]     **Huaizheng Zhang**
**Shudong Huang**[1]     **Qing Ye**[1†]     **Jiangcheng Lv**[1†]
[1]Sichuan University     [2]National University of Singapore

## Abstract

Classical federated learning (FL) enables training machine learning models without sharing data for privacy preservation, but heterogeneous data characteristic degrades the performance of the localized model. Personalized FL (PFL) addresses this by synthesizing personalized models from a global model via training on local data. Such a global model may overlook the specific information that the clients have been sampled. In this paper, we propose a novel scheme to inject personalized prior knowledge into the global model in each client, which attempts to mitigate the introduced incomplete information problem in PFL. At the heart of our proposed approach is a framework, the *PFL with Bregman Divergence* (pFedBreD), decoupling the personalized prior from the local objective function regularized by Bregman divergence for greater adaptability in personalized scenarios. We also relax the mirror descent (RMD) to extract the prior explicitly to provide optional strategies. Additionally, our pFedBreD is backed up by a convergence analysis. Sufficient experiments demonstrate that our method reaches the *state-of-the-art* performances on 5 datasets and outperforms other methods by up to 3.5% across 8 benchmarks. Extensive analyses verify the robustness and necessity of proposed designs. https://github.com/BDeMo/pFedBreD_public

## 1   Introduction

Federated learning (FL) [53] has achieved significant success in many fields [72, 43, 71, 76, 69, 59, 32, 5, 34, 49], which include recommendation systems utilized by e-commerce platforms [71], prophylactic maintenance for industrial machinery [76], disease prognosis employed in healthcare [69]. Data heterogeneity is a fundamental characteristic of FL, leading to challenges such as inconsistent training and testing data (data drift) [36]. An efficient solution to these challenges is to fine-tune the global model locally for adaptation on local data [3, 74, 35]. This solution is straightforward and pioneering, but presents a fundamental limitation when dealing with highly heterogeneous data. For examples, heterogeneous data drift may introduce substantial noise [29] and the resulted model may not generalize well to new sample [20, 12]. Thus, heterogeneous data in FL is still challenging [66].

Recently, personalized FL (PFL) is proposed to mitigate the aforementioned negative impact of heterogeneous data [66]. To improve the straightforward solution mentioned above, Per-FedAvg [20] is introduced to train a global model that is easier to fine-tune. Another paper on the similar topic, FedProx [45], aims to resolve the issue of personalized models drifting too far from the global model during training with a dynamic regularizer in the objective during local training. This issue could occur especially in post-training fine-tuning methods without regularization (*e.g.*, Per-FedAvg [20]). Moreover, pFedMe [65], another regularization method modeling local problems using Moreau

---

*3101ihs@gmail.com

†Equal Corresponding Authors {yeqing, lvjiancheng}@scu.edu.cn

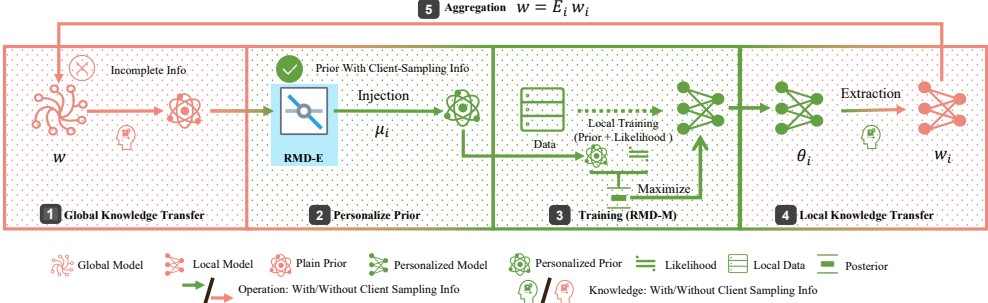

Figure 1: pFedBreD framework: Global-MLE and Local-MAP. The personalized prior knowledge is injected into the global model of the global problem (MLE) in the $2^{nd}$ step for local training. The local knowledge is extracted from the local problem (MAP) in the $4^{th}$ step for aggregation.

envelopes, replaces FedProx's personalized model aggregation method with an interpretable approach for aggregating local models [45]. It also accommodates first-order Per-FedAvg [20].

Although the existing PFL methods have achieved promising results, the prior knowledge from single global model for local training [66] hinders the development of PFL. Specifically, we analyze the shortcomings of current PFL methods as follows: 1) utilizing the same global model for direct local training could potentially disregard the client's sampling information. As shown in Figure 1, a single global model provides global knowledge directly for local training, which overlooks the client-sampling information when the global knowledge is transferred to specific clients. 2) Explicitly extracting prior knowledge can be a challenging task. Most of the insightful works [17, 52] propose assumptions for recovering this incomplete information, but these assumptions are implicit, which limits the way to use the information to develop personalized strategies.

To address the former issue above, we propose framework pFedBreD to inject personalized prior knowledge into the one provided by a global model. As shown in Figure 1, it is injected in the $2^{nd}$ step and the local knowledge is transferred into global model via local models instead of directly aggregating personalized models [45]. To address the latter, we introduce *relaxed mirror descent RMD* to explicitly extract the prior for exploring personalized strategies.

Our method is backed up with direct theoretical support from Bayesian modeling in Section 4 and a convergence analysis in Section 5, which provides a linear bound $\mathcal{O}(1/TN)$ with aggregation noise and a quadratic speedup $\mathcal{O}(1/(TNR)^2)$ without.[3] Meanwhile, the existence and validity of the injection and extraction aforementioned information is verified in Section 6.2. The remarkable performance of the implements of the proposed method is tested on 5 datasets and 8 benchmarks. Consistently, our method reach the *state-of-the-art*. Especially, the improvement of accuracy on task DNN-FEMNIST [11] is up to 3.5%. Extensive ablation study demonstrate that parts of the hybrid strategy **mh** are complementary to each other. Our contributions can be summarized as follows:

- The problem of overlooking client-sampling information at prior knowledge being transferred is introduced in this paper, and we first investigate the possibility of explicitly expressing the prior knowledge of the information and design personalized strategies on it.

- To express the personalized prior, we model PFL into a Bayesian optimization problem, *Global-MLE and Local-MAP*. A novel framework, pFedBreD, is proposed for computing the modeled problem, and RMD is introduced to explicitly extract prior information.

- Sufficient experiments demonstrate our method surpasses most baselines on public benchmarks, thereby showcasing its robustness to data heterogeneity, particularly in cases involving small aggregation ratios and non-convex local objective settings.

---

[3]$TN$ and $TNR$: total global / local epoch in FL system. See Appendix A.1.

## 2 Related Works

**Regularization**   Researchers have developed a variety of approaches based on regularization to handle the PFL challenge in recent years (*e.g.*, FedU [18], pFedMe [65], FedAMP [31], HeurFedAMP [31]). All of these approaches' personalized objective functions can be expressed as $J(\theta) + R(\theta; \mu)$ where $J(\theta)$ is the loss function of the local problem and $R(\theta; \mu)$ is the regularization term used to restrict the deviation between $\theta$ and $\mu$ (*e.g.*, $R(\theta; \mu) = \frac{1}{2}||\theta - \mu||^2$ in pFedMe).

**Meta Learning**   One of the most representative meta-learning based single-model PFL approach is the well-known Per-FedAvg [20], aiming to find an initialization that is easy to fine-tune. That is, the global model in the FL setting is regarded as a meta model in MAML [21, 19], where the objective function of the local problem is $J(\theta - \eta \nabla J(\theta))$. Researchers also show the connections between FedAvg [53] and Reptile [60], another meta learning framework. [35] shows how to improve personalization of FL via Reptile. Proximal updating is also used in meta-learning based algorithms such as [79]. One of our strategies **meg** is the one motivated by MAML.

**Expectation Maximization**   Two EM-based [15] methods are proposed, *e.g.*, FedSparse in [50] and FedEM in [17]. Both of them focus on communication compression. The latter provides a variance reduce version and assumes complete information (or data) of the global model obeys distribution in X-family. Another FedEM [52] combines Bayesian modeling, Federated Multi-task learning (FTML) and EM. Our framework pFedBreD is a expectation maximuzatioin and maximum a posteriori estimate (EM-MAP) [17] algorithm with personalized prior specified.

**Bayesian FL**   In recent years, studies of PFL with Bayesian learning have been proposed. In related approaches, FLOA [48] and pFedGP [1] are proposed with KL divergence regularization in the loss function, which is comparable to applying specific assumption of X-family prior in pFedBreD see Appendix B.2 for details. Our implementation doesn't use a Bayesian neural network (BNN) model as an inferential model as others do (*e.g.*, pFedGP uses a Gaussian process tree and pFedBayes [77] uses BNN). Instead, to eliminate weight sampling cost in Bayesian methods, prior knowledge is introduced through regularization term.

## 3 Preliminary

**Overlooked Information in Prior Knowledge**   From a Bayesian and info. perspective, the global knowledge transferred in conventional method with single global model has no mutual information (MI) with client sampling $i$, i.e., formally, $w = \mathbf{E}_i w_i = \mathbf{E}_i w_i | i \Rightarrow \text{MI } I(w; i) = 0$, in particular when applying reg. $R(w^{(t)}; ...)$ or local init. $w_{i,0}^{(t)} \leftarrow w$ where $w_i$ is the local model on the $i^{th}$ client. This makes the specific model on each client have to re-obtain this information from scratch solely from the data during training, especially impacted on hard-to-learn representations and datasets.

**Bregman-Moreau Envelope**   Bregman divergencee [10] is employed as a general regular term in our local objective that exactly satisfies the computational requirements and prior assumption, and is formally defined in Eq. (1).

$$\mathcal{D}_g(x, y) := g(x) - g(y) - \langle \nabla g(y), x - y \rangle \tag{1}$$

where $g$ is a convex function. For convenience, $g$ is assumed to be strictly convex, proper and differentiable such that Bregman divergence is well-defined. To utilize the computational properties of Bregman Divergence in optimization problems, we introduce the following definition in Eq. (2) [7, 8]: Bregman proximal mapping, Bregman-Moreau envelope, and the relationship between them.

$$\mathcal{D}\mathbf{prox}_{g,\lambda^{-1}} f(x) := \arg \min_\theta \{f(\theta) + \lambda \mathcal{D}_g(\theta, x)\},$$

$$\mathcal{D}\mathbf{env}_{g,\lambda^{-1}} f(x) := \min_\theta \{f(\theta) + \lambda \mathcal{D}_g(\theta, x)\}, \tag{2}$$

$$\nabla \mathcal{D}\mathbf{env}_{g,\lambda^{-1}} f(x) = \lambda \nabla^2 g(x)[x - \mathcal{D}\mathbf{prox}_{g,\lambda^{-1}} f(x)],$$

where $\lambda > 0$ denotes the regular intensity in general and the variance of the prior in our modeling.

**Exponential Family** The regular exponential family (X-family) is a relatively large family that facilitates calculations. Therefore, to yield the prior, we employ the X-family [6]defined in Eq. (3).

$$\mathbf{P}_{ef}(\mathcal{V}; s, g) = h(\mathcal{V}) \exp\{\langle \mathcal{V}, s \rangle - g(s)\} = h(\mathcal{V}) \exp\{-\mathcal{D}_{g^*}(\mathcal{V}, \mu) + g^*(\mathcal{V})\}, \tag{3}$$

where $g$ is assumed to be convex, $\mathcal{D}_g(\cdot, \cdot)$ is the Bregman divergence, and $g^*$ is the Fenchel Conjugate of $g$. In Eq. (3), $s$, $h(\mathcal{V})$ and $g(s)$ are respectively the natural parameter, potential measure and logarithmic normalization factor, where we have the mean parameter $\mu = \nabla g(s)$. Additionally, to highlight the variance, the scaled exponential family (SX-family) is introduced in Eq. (4)

$$\mathbf{P}_{sef}(\mathcal{V}; \lambda, s, g) = h_{\mathcal{V}}(\mathcal{V}) \exp\{\lambda[\langle \mathcal{V}, s \rangle - g(s)]\} = h_\lambda(\mathcal{V}) \exp\{-\lambda \mathcal{D}_{g^*}(\mathcal{V}, \mu) + \lambda g^*(\mathcal{V})\}, \tag{4}$$

where $\log h_\lambda(\mathcal{V})$ is the scaled potential measure, and the scale parameter $\lambda$ is employed to highlight the variance. Moreover, $\mathcal{V}$ is assumed to be the minimal sufficient statistic of the complete information for local inference, details of which can be found in Section 4.

## 4 Methodology

In this section[4], we introduce missing client-sampling information based on classic FL, use EM to reduce the computational cost of the information-introduced FL problem, and propose RMD, a class of prior selection strategies, based on the E-step in EM. The general FL classification problem with KL divergence could be formulated as Eq. (5) [53, 66].

$$\arg\min_w \mathbf{E}_i \mathbf{E}_{d_i} \mathbf{KL}(\mathbf{P}(y_i|x_i) || \hat{\mathbf{P}}(y_i|x_i, w)) = \arg\max_w \mathbf{E}_i \mathbf{E}_{d_i} \mathbf{E}_{y_i|x_i} \log \hat{\mathbf{P}}(y_i|x_i, w), (x_i, y_i) \in d_i, \tag{5}$$

where we rewrite the discriminant model as an maximum likelihood estimation (MLE) problem [41] of $y_i|x_i$ in the right hand side (R.H.S.) of Eq. (5). $(x_i, y_i)$ represent the pairs of input and label respectively in dataset $d_i$ on the $i^{th}$ client, and $\hat{\mathbf{P}}(y_i|x_i, w)$ is the inferential model parameterized by $w$. Each local data distribution is presuppose to be unique, so using the global model with local data for inference and training could overlook the fact that the client has been sampled before transmitting the global model, and the prior knowledge transmitted directly via the global model as the local training prior knowledge (*e.g.* via initial points, penalty points in dynamic regular terms, etc.) has no mutual information with the client sampling, *i.e.*, the global model $w = \mathbf{E}_i w_i = \mathbf{E}_i w_i | i$. Thus, to reduce the potential impact of the overlooked information, the complete information $\Theta_i$ on the $i^{th}$ client is introduced which turns Eq. (5) into Eq. (6).

$$\arg\max_w \mathbf{E} \log \int_{\Theta_i} \hat{\mathbf{P}}(y_i|x_i, \Theta_i, w) \mathbf{P}(\Theta_i|x_i, w) d\Theta_i, \tag{6}$$

where $\mathbf{E} = \mathbf{E}_i \mathbf{E}_{d_i} \mathbf{E}_{y_i|x_i}$ and the direct calculation of this is computationally expensive [14].

**Framework: Leveraging Expectation Maximization for Prior Parameter Extraction** The integral term in Eq. (6) makes direct computation impossible [14], so we employ EM to approximate the likelihood with unobserved variables [15] as shown in Eq. (7), where $\mathbf{Q}(\Theta_i)$ is any probability measure.

$$\sum_i \log \hat{\mathbf{P}}(y_i|x_i, w) \geq \sum_i \mathbf{E}_{\mathbf{Q}(\Theta_i)}[\log \hat{\mathbf{P}}(y_i|x_i, \Theta_i, w) + \mathbf{E}_{y_i|x_i, w} \log \mathbf{P}(\Theta_i|d_i, w)]. \tag{7}$$

Assuming that prior $\Theta_i|d_i, w \sim \hat{\mathbf{P}}_{sef}(\Theta_i; \lambda, s_i(w; d_i), g)$[5] and the local loss function on the $i^{th}$ client $f_i(\Theta_i, w)$ is $\mathbf{E}_{d_i}[-\log \mathbf{P}(y_i|x_i, \Theta_i, w)]$, we have the left hand side (L.H.S.) of the Eq. (8) from (7). Here is an assumption for simplification that $\theta_i$ contains all the information for local inference, *i.e.* $\theta_i = \Theta_i$ and $\mathbf{P}(y_i|x_i, \Theta_i, w) = \mathbf{P}(y_i|x_i, \Theta_i)$. It happens when $\theta_i$ is all the parameters of the personalized model and we only use the personalized model for inference. Thus, $f_i(\theta_i) = \mathbf{E}_{d_i}[-\log \hat{\mathbf{P}}(y_i|x_i, \theta_i)]$. Thus, we can optimize an upper bound as a bi-level optimization problem as shown in the R.H.S. of the Eq. (8) to solve Eq. (5) approximately, where mean parameter $\mu_i = \nabla g \circ s_i$[6] [9]. And, we can derivate our framework as shown in Section 5.

$$- \max_{w, \{\theta_i\}} \mathbf{E}_i\{-f_i(\theta_i) - \lambda \mathcal{D}_{g^*}(\theta_i, \mu_i(w))\} \leq \min_w \mathbf{E}_i \; \boxed{\min_{\{\theta_i\}}\{f_i(\theta_i) + \lambda \mathcal{D}_{g^*}(\theta_i, \mu_i(w))\}}. \tag{8}$$

---

[4]More details of equations are in Appendix B.

[5]A-posteriori distribution for local client whose prior knowledge is from global model. See Appendix B.

[6]The following $d_i$ is omitted with the same footnote $i$ in $\mu_i$ for simplification ($\mu_i(\cdot) \leftarrow \mu_i(\cdot; d_i)$).

**Strategies: Relaxing Mirror Descent for Prior Selection**     To extract the prior strategies and implement $\mu_i$ E-step of EM in close-form, we propose a method called relaxed mirror descent (RMD), where the mirror descent (MD) is EM in X-family [40]. MD can be generally written as Eq. (9) from the old $\hat{w}$ to the new one $\hat{w}^+$ in each iteration [54, 40].

$$\hat{w}^+ \leftarrow \arg\min_{\hat{\theta}}\{f(\hat{w}) + \langle\nabla f(\hat{w}), \hat{\theta} - \hat{w}\rangle + \hat{\lambda}\mathcal{D}_{\hat{g}}(\hat{\theta}, \hat{w})\}. \tag{9}$$

According to the Lagrangian dual, we rewrite the problem into a more general variant shown in Eq. (10) with relaxed restrictions and superfluous parameter.

$$\arg\min_{\hat{\theta}, \hat{\mu}}\{\Psi(\hat{\theta}, \hat{w}) + \langle\nabla\Phi(\hat{w}), \hat{\mu} - \hat{w}\rangle + \lambda\mathcal{D}_{g^*}(\hat{\theta}, \hat{\mu}) + (2\eta)^{-1}||\hat{\mu} - \hat{w}||^2\}. \tag{10}$$

We can transform Eq. (10) back into Eq. (9) by setting $\Phi(\hat{w})$ to satisfy $\nabla\Phi(\hat{w}) = \nabla f(\hat{w})$, and defining $\Psi(\hat{\theta}, \hat{w})$ as a function with $f(\hat{w})$ and a penalty term to make $\hat{\theta}$ and $\hat{w}$ close as possible (*e.g.*, $\hat{\lambda}\mathcal{D}_{\hat{g}}(\hat{\theta}, \hat{w})$). This provides us a way to extract $\mu_\Phi$ the function to generate mean parameter of the prior, as shown in Eq. (11), which is minimizing an upper bound of the problem in Eq. (10).

$$\mathcal{D}\mathbf{env}_{g^*, \lambda^{-1}}\Psi(\cdot, w)(\mu_\Phi(w)) = \min_{\theta}\{\Psi(\theta, w) + \lambda\mathcal{D}_{g^*}(\theta, \mu_\Phi(w))\}$$
$$\mu_\Phi(w) = \arg\min_{\mu}\{\langle\nabla\Phi(w), \mu - w\rangle + (2\eta)^{-1}||\mu - w||^2\}. \tag{11}$$

By optimality condition, we have $\mu_\Phi(w) = w - \eta\nabla\Phi(w)$, which can be specified by $\Phi$. The remaining part is a Bregman-Moreau envelope. Thus, we can optimize the upper bound with an EM-MAP method, alternately computing $\mu_\Phi(w)$ and $\mathcal{D}\mathbf{prox}_{g^*, \lambda^{-1}}\Psi(\cdot, w)(\mu_\Phi(w))$.

# 5   Framework Design

**Problem Formulation that Highlights Personalized Prior**     Inspired by the aforementioned motivation, the personalized models $\theta_i$ and mean parameters are respectively the solution of $\mathcal{D}\mathbf{env}_{g^*, \lambda^{-1}}f_i(\mu_i(w))$ and $\mu_i(w)$ on the $i^{th}$ client, where $w$ is the global model. We assume that personalized model contains all the local information required for inference on the $i^{th}$ client, and satisfies $\theta_i|d_i, w \sim \mathbf{P}_{sef}(\theta_i; \lambda, s_i(w), g)$. The global problem can be written as Eq. (12).

$$\min_{w}\mathbf{E}_i\{F_i(w) := \mathcal{D}\mathbf{env}_{g^*, \lambda^{-1}}f_i(\mu_i(w))\}. \tag{12}$$

The given $g$ is strictly convex, $\lambda > 0$, $f_i$ is the local loss function, $s_i(w)$ is the natural parameter and $\mu_i(w) = \mathbf{E}_{\theta_i|x_i, w}\theta_i = \nabla g(s_i(w))$ is the mean (or expectation) parameter in Eq. (12).

**Framework: pFedBreD**     To solve the optimization problem in Eq. (12), we use gradient-based methods to solve the global problem using the gradient of $F_i$:

$$\nabla F_i(w) = \lambda\mathbf{D}\mu_i(w)\nabla^2 g^*(\mu_i(w))[\mu_i(w) - \mathcal{D}\mathbf{prox}_{g^*, \lambda^{-1}}f_i(\mu_i(w))], \tag{13}$$

where $\mathbf{D}$ is the gradient operator of the vector value function, and $\nabla^2$ is the Hessian operator.[7] The framework is shown as Algorithm 1, where $\mathcal{I}$ is the client selecting strategy for global model aggregation; $w_{init}$ and $\theta_{init}$ are the initialization strategies on the $i^{th}$ client; $\alpha_m$ is the main problem step-size; $T, R, N$ are respectively the total number of iterations, local iterations, and clients. $\beta$ is used in the same trick as [37, 65]. The strategies to derive the initialization points of $w_i$ and $\theta_i$ at each local epoch are $w_{i,0}^{(t)} \leftarrow w^{(t-1)}$ and $\theta_{i,0}^{(t)} \leftarrow \theta_{i,R}^{(t-1)}$.

**Implementation: Maximum Entropy and Meta-Step**     Practically, two main parts of the pFedBreD are needed to be implemented:

- $g$, the function used to derive the logarithmic normalization factor, determines the type of prior to be used;
- $\{s_i\}$ or $\{\mu_i\}$, the functions used to derive the natural parameter and mean parameter for the personalized local prior, determine which particular prior is used.

---

[7]The details of first-order methods is in Appendix A.7.

---

**Algorithm 1** Algorithm for pFedBreD

---

**Input**: $\mathcal{I}, \{d_i\}, i = 1...N$
**Parameter**: $\alpha_m, g, \lambda, T, R, \{w_{init}, \theta_i, \mu_i, \}, i = 1...N$
**Output**: $w^{(T)}, \{\theta_i^{(T)}\}, i = 1...N$

1:  Initialize $w^{(0)}, \{\theta_i^{(0)}\}, \{\mathcal{C}_{i,R}^0\}$;
2: **for** t=1...T **do**
3:      Server sends $w^{(t-1)}$ to clients ;
4:      **for** i=1...N in parallel on each clients **do**
5:          Initialize $w_{i,0}^{(t)}$ and $\theta_{i,0}^{(t)}$ with $w_{init}$ and $\theta_{init}$;
6:          **for** r=1...R **do**
7:              Generate $\mu_{i,r}^{(t)} \leftarrow \mu_i(w_{i,r-1}^{(t)}, ...)$ ;
8:              $\theta_{i,r}^{(t)} \leftarrow \mathcal{D}\mathbf{prox}_{g^*, \lambda^{-1}} f_i(\mu_{i,r}^{(t)})$ ;
9:              $w_{i,r}^{(t)} \leftarrow w_{i,r-1}^{(t)} - \alpha_m \nabla F_i(w_{i,r-1}^{(t)})$ ;
10:          **end for**
11:      **end for**
12:      Server collects $\{w_{i,R}^{(t)}\}$ and calculate $w^{(t)} \leftarrow (1-\beta)w^{(t-1)} + \beta \mathbf{E}_\mathcal{I} w_{i,R}^{(t)}$ ;
13: **end for**
14: **return** $w^T, \{\theta_i^T\}$.

---

We propose first-order implementations based on maximum entropy rule [22, 33]. In the SX-family, the Gaussian distribution has the maximum entropy among continuous distributions when $g$, $\mu_i$ (the first-order moment), and $\lambda$ (the parameter determining the second moment) are given. Thus, we employ the scaled norm square $g = g^* = \frac{1}{2}||\cdot||^2$ to turn the prior into a spherical Gaussian, in order to maximize the entropy of the prior on a particular client. With this assumed prior, we have $\nabla g = \nabla g^* = I$, which means $\mu_i = s_i$. We can choose a different $\Phi_i$ as shown in Eq. (14)[8] to generate selection strategies according to Section 4, via $\mu_i(w) = w - \eta \nabla \Phi_i(w)$ (meta-step).

$$\Phi_i = \begin{cases} f_i \\ F_i \\ f_i + F_i \end{cases} \qquad \mu_{i,r}^{(t)} \leftarrow \begin{cases} w_{i,r-1}^{(t)} - \eta_\alpha \nabla f_i(w_{i,r-1}^{(t)}), & \mathbf{lg} \\ w_{i,r-1}^{(t)} - \eta(w_{i,R}^{(t-1)} - \theta_{i,r-1}^{(t)}), & \mathbf{meg} \\ w_{i,r-1}^{(t)} - \eta_\alpha \nabla f_i(w_{i,r-1}^{(t)}) - \eta(w_{i,R}^{(t-1)} - \theta_{i,r-1}^{(t)}), & \mathbf{mh} \end{cases} \quad (14)$$

where $\eta_\alpha$ and $\eta$ are the meta-step-size parameters. Practical parameter selection strategies with meta-step are shown as $\mu_{i,r}^{(t)}$ in Eq. (14). The three of $\mu_i$, *i.e.* **lg**, **meg** and **mh**, represent **loss gradient**, **memorized envelope gradient** and **memorized hybrid** respectively.

**Convergence Analysis**    we analyze the convergence of pFedBreD with RMD on a uniform client sampling $\mathbf{E}_i = \frac{1}{N}\sum_{i=1}^N$ setting for simplification. Other sampling methods can be obtained with client sampling expectation $\mathbf{E}_i[F_i] = F$, by changing sampling weights. The assumptions, proof sketch and detailed notations are in Appendix A.1 and Appendix D.

**Theorem 1** (pFedBreD's global bound)*. Under settings in Section 5 and Appendix D, at global epoch $T \geq \frac{2}{\hat{\mu}_{F.}\tilde{\alpha}}$, by properly choose $\tilde{\alpha}_m = \alpha_m \beta R, \exists \tilde{\alpha}_m \leq \min\{\frac{\beta}{\sqrt{2\hat{c}}}, \frac{2}{\hat{\mu}_{F.}}, \hat{\alpha}_m\}$, where $A = [\frac{\hat{L}_{g^*}}{\hat{\mu}_{F.}}(\hat{u}_m + \eta\hat{\gamma}_\Phi)]^2(\frac{\hat{\gamma}_f^2}{|\hat{d}_i|} + \hat{\epsilon}^2)$, $B = [\hat{L}_\mathcal{E}\hat{\gamma}_\Phi(1 + \sigma_\Phi)(\hat{u}_m + \eta\hat{\gamma}_\Phi)]^2$, $C = \frac{\sigma_\Phi^2 \hat{L}_\mathcal{E}^2(\hat{u}_m + \eta\hat{\gamma}_\Phi)^2}{\hat{\mu}_{F.}^3}$, $\xi^{(t)} = (1 - \frac{\tilde{\alpha}\hat{\mu}_{F.}}{2})^{-t-1}$, $\bar{w}^{(T)} := \frac{\sum_{t=0}^{T-1} \xi^{(t)} w^{(t)}}{\sum_{t=0}^{T-1} \xi^{(t)}}$ and $\hat{\alpha}_m := \frac{\hat{\mu}_{F.}\beta R}{e(1 + \sigma_\Phi)\hat{L}_\mathcal{E}(\hat{u}_m + \eta\hat{\gamma}_\Phi)2^{R + 6}\frac{1}{2}(\frac{1}{R} + 2) + 18(\hat{\mu}_{F.}\beta R)\hat{L}_F}$, such that:*

$$\mathcal{O}[\mathcal{D}_F(\bar{w}^{(T)}, w^*)] = \mathcal{O}(\hat{\mu}_{F.}e^{-\tilde{\alpha}_m\hat{\mu}_{F.}T/2}\mathbf{\Delta}^{(0)}) + \mathcal{O}(\frac{A\lambda^2 + B}{\hat{\mu}_{F.}})$$
$$+ \mathcal{O}(\frac{(N/S - 1)\sigma_{F,*}{}^2}{NT\hat{\mu}_{F.}}) + \mathcal{O}(\frac{2^R C}{T^2\beta^2 R^2}[R\sigma_{F,*}^2 + A\lambda^2 + B]).$$

---

[8]A variant of **mh** is in Appendix C.2.

.

**Theorem 2** (pFedBreD$_{ns}$'s first-order personalization bound). *Under the same conditions as in Theorem 1, with prior assumption of a spherical Gaussian and first-order approximation, the bound for the gap between the personalized approximate model and global model in the Euclidean space is:*

$$\mathbf{E}||\tilde{\theta}_i(\bar{w}^T) - w^*||^2 \leq \mathcal{O}(\dot{\delta}_p) + \mathcal{O}[\dot{c}_p \mathcal{D}_F(\bar{w}^{(T)}, w^*)]$$

*where* $\dot{\delta}_p = \frac{2}{\hat{\mu}_{F_{i,\cdot}}^2}(\frac{\hat{\gamma}_f^2}{|\dot{d}_i|} + \hat{\epsilon}^2) + \frac{2}{\lambda^2}\epsilon_1^2 + \frac{4}{\lambda^2}\sigma_{F,*}^2 + \frac{1}{2}\eta^2\mathcal{G}_\Phi^2$, *and* $\dot{c}_p = (\frac{32}{\lambda^2}\hat{L}_F + \frac{8}{\hat{\mu}_{F\cdot}})$.

**Remark 1.** *Theorem 1 shows the main factors that affect the convergence of a global model are as follows: random mini-batch size, client drift error, aggregation error, heterogeneous data, dual space selection, local approximation error, and selection strategy for exponential family prior mean and variance. These can be divided into four categories based on their computational complexity. The first and second term shows that the proper fixed $\tilde{\alpha}_m$ can linearly reduce the influence of initial error $\mathbf{\Delta}^{(0)}$ and the global model converges to a ball near the optimal point. The radius of this ball is determined by the personalized strategy and local errors (including local data randomness and envelope approximation errors). The third term implies that a linear convergence rate $\mathcal{O}(1/(NT))$ can be obtained w.r.t. the total global epoch $NT$ in the presence of aggregation noise. Without client sampling $N = S$, according to the fourth term, the quadratic rate $\mathcal{O}(1/(TNR)^2)$ can be obtained with $\beta = \mathcal{O}(N)$ or $\beta = \mathcal{O}(N\sqrt{R})$ (Note that the number of local epoch $R$ cannot be too large due to client drift, according to $2^R$). Theorem 2 shows that, with spherical Gaussian prior assumption and first-order methods, the radius of the neighborhood range for the minimum that includes the personalized model on $i^{th}$ client, $\mathcal{O}(C_{\Phi,F,f,d} + \frac{1}{\lambda^2}(\epsilon_1^2 + \sigma_{F,*}^2 + \frac{B\hat{L}_F}{\hat{\mu}_{F\cdot}}) + \lambda^2\frac{A}{\hat{\mu}_{F\cdot}})$, can be trade-off by $\lambda$, and is affected by the prior selection strategies and first-order approximate error besides the elements in Theorem 1. (Note that the Euclidean space is self-dual.)*

# 6 Experiments

## 6.1 General Settings

**Tricks, Datasets and Models:** our experiments include several tasks: CNN [28] on CIFAR-10 [18, 39], LSTM [27] on Sent140 [11] and MCLR/DNN on FEMNIST [11]/FMNIST [65, 67]/MNIST[65, 42]. The details of tricks (FT, AM), data heterogeneity and models are in Appendix C.

**Baselines:** we choose following algorithms as our baselines: FedAvg [53], Per-FedAvg [20], pFedMe [65], FedAMP [31], pFedBayes [77] and FedEM [52]. These baselines are respectively classical FL, MAML-based meta learning, regularization based, FTML methods, variational inference PFL and FMTL with EM.

**Global Test and Local Test:** the global and personalized model, represented by **G** and **P**, are evaluated with global and local tests respectively. Global test means all the test data is used in the test. Local test means only the local data is used for the local test and the weight of the sum in local test is the ratio of the number of data. The results of average accuracy per client are shown in Table 1. Each experiment is repeated 5 times. More details are in Appendix C. For readability, we only give the error bar in the main Table 1 and Table 2, and keep one decimal except for the main Table 1.

**Hyperparameter Settings** The step-size of the main problem, $\alpha_m$, and the personalized step-size, $\alpha$, for all methods are 0.01. $\beta$ is 1, and the number of local epochs, $R$, is 20 for all datasets. $\lambda$ is chosen from 15.0 to 60.0. The batch sizes of Sent140 and the other datasets are 400 and 20, respectively, and the aggregation strategy, $\mathcal{I}$, is uniform sampling. The ratios of aggregated clients per global epoch are 40%, 10%, and 20% for Sent140, FEMNIST, and the other datasets, respectively. The numbers of total clients, $N$, are 10, 198, 20, and 100 for Sent140, FEMNIST, CIFAR-10, and other datasets. The number of proximal iterations is 5 for all settings with proximal mapping. In our implementations, $\eta_\alpha$ and $\eta$ are respectively 0.01 and 0.05.

**Summarizing the Effects of Hyper-parameters** We test the hyper-parameter effect of $\eta$ and $\lambda$ in our implementation pFedBreD$_{ns,\mathbf{mh}}$. The details are in Appendix C. From the results, we find that it will degrade the test accuracy if the values of $\lambda$ or $\eta$ are too large or too small. The test accuracy of personalized model is more sensitive than the ones of global model. The test accuracy of personalized

Table 1: Results of average testing accuracy (%) per client of each settings. We mark the best and second best performance by **bold** and underline. Avg and Std : the average results and the standard deviation of them on all tasks; H.Avg and H.Std : the average results and the standard deviation of them on hard tasks (non-linear DNN with complex classification or architecture: DNN / CNN / LSTM on FEMNIST / CIFAR-10 / Sent140). The **G** and **P** are global and personalized model

| Methods / Datasets | FEMNIST | | FMNIST | | MNIST | | CIFAR-10 | Sent140 | Statistics | | | |
| Names - **Models** | MCLR | DNN | MCLR | DNN | MCLR | DNN | CNN | LSTM | Avg | Std | H.Avg | H.Std |
|---|---|---|---|---|---|---|---|---|---|---|---|---|
| FedAvg [53] -G | $53.38_{\pm0.26}$ | $57.04_{\pm0.08}$ | $82.75_{\pm0.04}$ | $80.09_{\pm0.06}$ | $86.59_{\pm0.03}$ | $88.26_{\pm0.05}$ | $57.51_{\pm0.07}$ | $70.86_{\pm0.01}$ | 72.06 | 14.34 | 61.80 | 7.85 |
| FedAvg+AM -G | $55.34_{\pm0.05}$ | $59.03_{\pm0.10}$ | $82.58_{\pm0.03}$ | $81.03_{\pm0.12}$ | $86.74_{\pm0.03}$ | $89.31_{\pm0.05}$ | $57.07_{\pm0.12}$ | $71.27_{\pm0.01}$ | 72.80 | 14.01 | 62.46 | 7.70 |
| FedEM [52] -G | $40.75_{\pm0.32}$ | $45.47_{\pm0.04}$ | $95.78_{\pm0.03}$ | $96.42_{\pm0.03}$ | $85.75_{\pm0.01}$ | $86.49_{\pm0.02}$ | $57.67_{\pm0.16}$ | $66.72_{\pm0.03}$ | 71.88 | 22.28 | 56.62 | 10.66 |
| pFedBayes [77] -P | $49.66_{\pm0.46}$ | - | $98.46_{\pm0.05}$ | $98.67_{\pm0.05}$ | $89.64_{\pm0.06}$ | $90.48_{\pm0.12}$ | - | - | - | - | - | - |
| FedAMP [31] -P | $60.04_{\pm0.08}$ | $66.79_{\pm0.04}$ | **$98.63_{\pm0.02}$** | $98.72_{\pm0.01}$ | **$90.81_{\pm0.02}$** | $92.21_{\pm0.02}$ | $77.40_{\pm0.04}$ | $69.83_{\pm0.05}$ | 81.80 | 15.21 | 71.34 | 5.46 |
| pFedMe [65] -P | $50.74_{\pm0.10}$ | $53.56_{\pm0.12}$ | $97.60_{\pm0.03}$ | $98.63_{\pm0.01}$ | $88.20_{\pm0.05}$ | $90.51_{\pm0.01}$ | $72.24_{\pm0.05}$ | $69.36_{\pm0.02}$ | 77.61 | 18.96 | 65.05 | 10.06 |
| pFedMe+FT -P | $58.04_{\pm0.11}$ | $62.93_{\pm0.10}$ | $97.63_{\pm0.01}$ | $98.39_{\pm0.02}$ | $88.36_{\pm0.02}$ | $91.71_{\pm0.01}$ | $68.17_{\pm0.11}$ | $67.82_{\pm0.03}$ | 79.13 | 16.53 | 66.31 | **2.93** |
| pFedMe+AM -P | $55.56_{\pm0.09}$ | $60.08_{\pm0.05}$ | $97.57_{\pm0.02}$ | $98.67_{\pm0.00}$ | $88.46_{\pm0.02}$ | $91.22_{\pm0.00}$ | $73.35_{\pm0.09}$ | $70.93_{\pm0.05}$ | 79.48 | 16.79 | 68.12 | 7.07 |
| Per-FedAvg [20] -P | $54.34_{\pm0.14}$ | $62.72_{\pm0.03}$ | $94.28_{\pm0.05}$ | $97.46_{\pm0.04}$ | $87.09_{\pm0.01}$ | $90.96_{\pm0.02}$ | $78.87_{\pm0.05}$ | $70.05_{\pm0.03}$ | 79.47 | 15.74 | 70.54 | 8.09 |
| Per-FedAvg+FT -P | $55.34_{\pm0.15}$ | $63.34_{\pm0.01}$ | $95.76_{\pm0.07}$ | $98.10_{\pm0.01}$ | $87.56_{\pm0.03}$ | $89.58_{\pm0.01}$ | $79.68_{\pm0.04}$ | $70.20_{\pm0.01}$ | 79.95 | 15.61 | 71.07 | 8.20 |
| Per-FedAvg+AM -P | $56.66_{\pm0.09}$ | $65.74_{\pm0.02}$ | $92.08_{\pm0.10}$ | $98.24_{\pm0.02}$ | $86.91_{\pm0.04}$ | $90.85_{\pm0.02}$ | $78.97_{\pm0.03}$ | $70.73_{\pm0.05}$ | 80.02 | 14.54 | 71.81 | 6.68 |
| mh (ours) -P | $56.34_{\pm0.09}$ | $64.93_{\pm0.03}$ | $98.44_{\pm0.01}$ | $98.73_{\pm0.01}$ | $89.83_{\pm0.02}$ | $92.04_{\pm0.01}$ | $\underline{79.44}_{\pm0.02}$ | $\underline{72.04}_{\pm0.01}$ | 81.47 | 15.88 | 72.14 | 7.26 |
| mh (ours)+FT -P | $59.81_{\pm0.07}$ | $\underline{67.53}_{\pm0.02}$ | $\underline{98.51}_{\pm0.02}$ | **$98.98_{\pm0.03}$** | $\underline{90.10}_{\pm0.03}$ | **$92.96_{\pm0.05}$** | $79.16_{\pm0.03}$ | $71.87_{\pm0.01}$ | $\underline{82.37}$ | 14.92 | $\underline{72.85}$ | 5.88 |
| mh (ours)+AM -P | **$60.64_{\pm0.02}$** | **$70.34_{\pm0.01}$** | $98.48_{\pm0.01}$ | $\underline{98.75}_{\pm0.01}$ | $89.88_{\pm0.01}$ | $92.32_{\pm0.01}$ | **$80.60_{\pm0.01}$** | **$73.68_{\pm0.01}$** | **83.09** | **14.01** | **74.87** | $\underline{5.23}$ |

model is more sensitive to $\eta$ than to $\lambda$. Note that the hyper-parameters are roughly tuned, which shows the insensitivity of **mh**, and better tuning could improve the performance in the Table 1.

## 6.2 Analysis

**Comparative Analysis of Performance**   We compare our methods and the baselines from different perspectives, including convex or non-convex problems, easy or hard tasks, and text tasks. Additionally, we briefly discuss the absence of BNN on hard tasks.

**Convex or non-convex:** on non-convex problems, especially in hard tasks, our method significantly outperforms other methods by at least 3.06% employing some simple tricks. On convex problem, FedAMP outperforms our method somewhat on convex problems with simple data sets. One explanation is that the learning lanscape is simple in shape for these problems and FedAMP converges faster for this case. One possible reason for this is that since FedAMP uses the distance between models as a similarity in the penalty point selection, giving greater weight to the model that is most similar to the local one. In the later stages of training, since there is only one global optimum, this penalty point tends not to change, and thus the method degenerates into a non-dynamic regular term. Compounding intuition, this method will not be as advantageous for non-convex problems and harder convex problems, as penalty point tends to fall into the local optimum and lead to degradation of the dynamic regular term.

**From easy to difficult task:** from the difference between the statistics of Avg and H.Avg in Table 1, it can be observed that meta-step methods perform most consistently, with all other methods dropping at least 10%. This is due to the simple and effective local loss design of MAML, with its learning-to-learning design philosophy that enables the method to be more stable in complex situations [19, 20].

**Personalized prior on text:** text tasks, as opposed to image tasks, generally have relatively rugged learning landscape. [55, 16, 13] This understanding is manifested in specific ways, such as parameter sensitivity, slow convergence, and struggling during the process. Thus, the overlooked prior information seems to be more important, which means that each local iteration not only obtains local knowledge from the data, but also the prior itself already contains some local knowledge. Therefore, there is no need to re-obtain this knowledge from scratch solely from the data during training.

**Absence of BNN on hard tasks:** complex BNN is not in Table 1, such as LSTM in pFedBayes, because it is difficult to conduct comparative experiments by fixing elements, *e.g.*, inferential models, tricks and optimization methods. In pFedBayes, training often crashes on hard tasks and large datasets, as mentioned in [77]. Our one-step-further research shows that it may be caused by the reparameterization tricks and vanilla Gaussian sampling. If we add tricks on it, the implementation will be very different from the original pFedBayes, and it is beyond this analysis.

**Ablation Analysis of Personalized Prior**   We conduct ablation experiments by dropping the gradient of the Bregman-Moreau envelope, the local loss function, or both, from the personalized

Table 2: Average local test accuracy of personalized model (%) in ablation experiments.(↑/↓: average accuracy is increased/reduced; AC4PP: Additional cost for personalized prior; Grad. and Add.: cost about calculate gradient and addition; Other notations are the same in Table 1.)

| Methods | FEMNIST | | FMNIST | | MNIST | | CIFAR-10 | Sent140 | Statistics | | AC4PP |
|---|---|---|---|---|---|---|---|---|---|---|---|
| | MCLR | DNN | MCLR | DNN | MCLR | DNN | CNN | LSTM | Avg | H.Avg | |
| Non-PP | $50.7_{\pm0.10}$ | $53.6_{\pm0.12}$ | $97.6_{\pm0.03}$ | $98.6_{\pm0.01}$ | $88.2_{\pm0.05}$ | $90.5_{\pm0.01}$ | $72.2_{\pm0.05}$ | $69.4_{\pm0.02}$ | 77.6 | 65.1 | None |
| lg (ours) | $50.8_{\pm0.05}$ | $49.1_{\pm0.53}$ | $98.3_{\pm0.02}$ | $98.4_{\pm0.02}$ | $88.4_{\pm0.01}$ | $91.0_{\pm0.00}$ | $65.7_{\pm0.46}$ | $60.7_{\pm0.41}$ | 75.3↓ | 58.5↓ | Grad. × R |
| meg (ours) | $50.3_{\pm0.07}$ | $53.9_{\pm0.06}$ | $97.8_{\pm0.00}$ | $98.6_{\pm0.01}$ | $88.4_{\pm0.01}$ | $90.6_{\pm0.01}$ | $73.8_{\pm0.06}$ | $69.4_{\pm0.02}$ | 77.9↑ | 65.7↑ | Add. × R |
| mh (ours) | $56.3_{\pm0.09}$ | $64.9_{\pm0.03}$ | $98.4_{\pm0.01}$ | $98.7_{\pm0.01}$ | $89.8_{\pm0.02}$ | $92.0_{\pm0.01}$ | $79.4_{\pm0.02}$ | $72.0_{\pm0.01}$ | 81.5↑ | 72.1↑ | Both above |

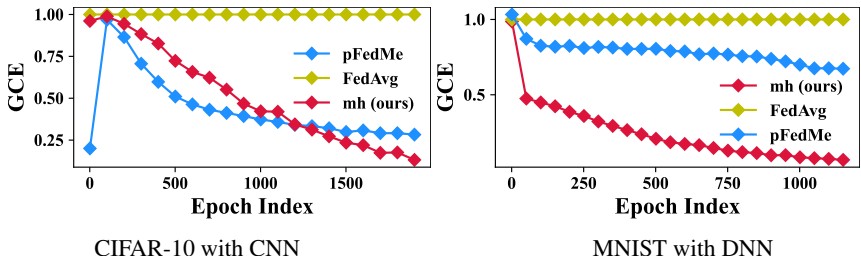

CIFAR-10 with CNN      MNIST with DNN

Figure 2: The results of GCE($\{\nabla F_i(w_i^{(t)})\}$) at each global epoch $t$ after Savitzky-Golay filtering [62].

strategy **mh** as shown in Table 2. The relationship among the three strategies mentioned in Eq. (14) is that **mh** consists of **lg** and **meg**. Moreover, pFedMe can be regarded in our framework as the one which takes the spherical Gaussian as prior and uses vanilla prior selection strategy $\mu_i = I$ without personalization. Thus, pFedMe and the three implementations of pFedBreD are compared. The results reveal the instability of our implementation **lg** and the introduction of **meg** on difficult tasks is about the same as not introducing it. However, introducing both **lg** and **meg** (i.e., **mh**) together shows remarkable performance. This indicates that **lg** and **meg** complement each other. **To explain these results**, by observing the error bars, in most of the settings, **meg** is significantly more stable compared to methods that do not use personalized priors, while **lg** is relatively less stable. Based on this observation, we have reason to believe that **meg** weakens the influence of potential noise, while **lg** introduces new noise. Therefore, we can infer that while the mean parameters are steadily biased towards the personalized model, the introduction of new noise finds a path that is more likely to escape from local optima or saddle points, based on implicit regularization [61, 58, 57].

**Generalized Coherence Analysis of Information Injection and Extraction** The generalized coherence estimate (GCE) [25] of vectors from personalized to local model (*i.e.*, the envelope gradients in pFedMe and ours) among clients on each global epoch are shown in Figure 2. The smaller the GCE, the less coherent the envelope gradient between individual nodes and the greater the diversity of information in the global model update. As shown in Figure 2, we can observe that during the convergence phase, using a personalized prior method has significantly greater information diversity than not using a personalized prior method, which proves the success of injecting personalized prior knowledge into the global model and extracting local knowledge from the local training.

**Variable-Control Analysis of Robustness** We analyze the impact of aggregation noise and data heterogeneity [29] on our method, mainly **mh**, by controlling variables. Results are in Table 3 and Table 4. (Details are in Appendix C.8.) We test the performance of global model on different aggregation ratios, where all hyper-parameters except for the aggregation ratios are fixed. Meanwhile, we test the performance of both global and the personalized model on different data heterogeneity settings, where full aggregation (sample client equals total number of clients, $S = N$) and one-step local update (local epoch $R = 1$) are employed to get rid of the effects of aggregation noise and client drift. The experiments demonstrate the instability of the global model in **mh** at small aggregation ratios, which most of the other PFL methods have, by comparing their performances on different aggregation numbers. Comparing to the baselines, the experiments also demonstrate the relative robustness of our method to extreme data heterogeneity.

Table 3: The global test accuracy (%) of the global model with different numbers of clients for aggregation $S \in \{10, 20, 50, 100\}$.(♠:FEMNIST, ◇:FMNIST)

| Numbers | small | | ⟶ | large | Std |
|---|---|---|---|---|---|
| ♠-DNN | 59.0 | 60.1 | 60.1 | 59.8 | 0.5 |
| ♠-MCLR | 54.4 | 55.4 | 55.4 | 55.5 | 0.5 |
| ◇-DNN | 75.1 | 79.6 | 79.4 | 79.3 | 2.2 |
| ◇-MCLR | 80.0 | 82.6 | 81.8 | 82.7 | 1.3 |

Table 4: The local test accuracy (%) of the personalized model on FMNIST-DNN setting with different data heterogeneity (Non-IID) settings $\alpha \in \{0.01, 0.1, 1, 10, 100, 1000\}$($\alpha \downarrow$, Non-IID↑) [29]. The **Bolded** means the best.

| Non-IID | small | | ⟶ | | | large | Avg |
|---|---|---|---|---|---|---|---|
| FedAvg-**G** | 18.2 | 14.8 | 14.5 | 11.9 | 11.3 | 11.2 | 13.7 |
| pFedMe-**P** | 89.5 | 58.2 | 24.2 | 12.3 | 11.8 | 10.6 | 34.4 |
| pFedMe-**G** | 17.0 | 14.3 | 14.1 | 12.3 | 10.8 | 10.9 | 13.2 |
| **mh**(ours)-**P** | **89.6** | **58.7** | **25.2** | **13.1** | 11.1 | 11.0 | **34.8** |
| **mh**(ours)-**G** | 17.1 | 14.6 | 14.6 | 12.4 | **11.9** | **11.9** | 13.8 |

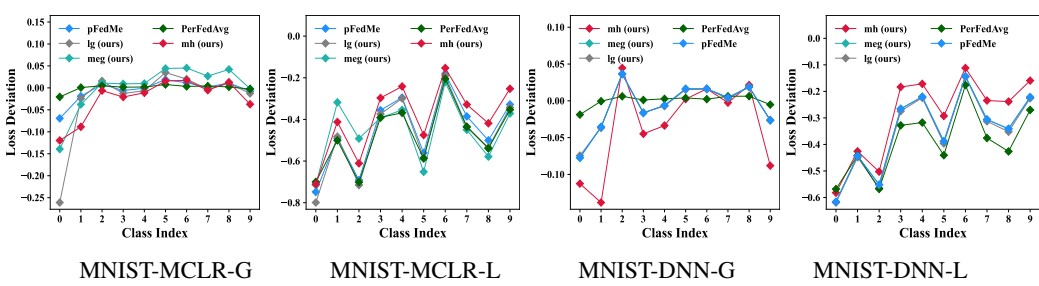

MNIST-MCLR-G      MNIST-MCLR-L      MNIST-DNN-G      MNIST-DNN-L

Figure 3: The loss deviation of experiments in Section 6 on the first client, whose major data are on $0^{th}$ classes. The lower deviation of the available class on global tests and the higher deviation of the unavailable class on local tests demonstrate the superior personalization ability of our methods.

**Deviation Analysis of Personalization**   Deviation represents the difference between an individual and the mean value. We use the deviation of the loss function to reflect the personalization. **On global test**, the lower the deviation, the better the personalized model performance on the corresponding local data. **On local test**, the model is only tested on its own dataset, and because of multiple local iterations, the local test deviation converges to almost the same value, as shown in MNIST-MCLR-L and MNIST-DNN-L in Figure 3. Furthermore, since the local test has a loss of 0 on missing classes, a higher deviation on missing classes reflects a lower mean on these classes. Thus, the lower loss in local testing and better performance are reflected from both of the almost equal deviation in local testing and the higher deviation on missing class. **Summary:** based on Figure 3, we can see that our method has higher deviation on missing classes in local testing and lower deviation in global testing. This means that our method has better personalized performance.

## 7 Conclusion and Discussion

**Conclusion**   To address the issue of neglecting client-sampling information while providing prior knowledge to local training via direct use of a global model, we propose a general concept: the personalized prior. In this paper, we propose a general framework, pFedBreD, for exploring PFL strategies under the SX-family prior assumption and computation, the RMD to explicitly extract the prior information, and three optional meta-step strategies to personalize the prior. We analyze our proposal both theoretically and empirically. Our strategy **mh** shows remarkable improvement in personalization and robustness to data heterogeneity on non-i.i.d. datasets and the LEAF benchmark [11] with MCLR / DNN / CNN / LSTM as inferential model, which conduct convex / non-convex problems, and image / language benchmarks.

**Limitations and Future Work**   Although **mh** shows remarkable performance and robustness, there is still instability in the global model with aggregation noise. Furthermore, it should be noted that the superficial reason for the improvement of **mh** seems to be that $\eta_\alpha$ and $\eta$ and (which are similar to each other) are used simultaneously, resulting in a magnitude in **mh** that is twice as large as the ones in the other two implementations and leading to better performance. However, empirically, simply doubling $\eta_\alpha$ in **lg** or $\eta$ in **meg** does not improve performance, and using one more **meg** step used in **lg** significant improvement. Our theoretical analysis cannot explain this phenomenon, and more detailed modeling is needed.

## Acknowledge

This work is supported by the Key Program of National Science Foundation of China (Grant No. 61836006) from College of Computer Science (Sichuan University) and Engineering Research Center of Machine Learning and Industry Intelligence (Ministry of Education), Chengdu 610065, P. R. China. This research is also supported by the National Research Foundation, Singapore under its AI Singapore Programme (AISG Award No: AISG2-PhD-2021-08-008).

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
