# Supplementary of PRIOR: Personalized Prior for Reactivating the Information Overlooked in Federated Learning

## A  Glossary, Some Basic Knowledge and Details about Implementations

### A.1  Glossary

The main notations in this paper are shown in Table 5.

Table 5: The glossary of notations mentioned in this paper

| Notation | Implication |
|---|---|
| $\cdot_i$ | $\cdot$ on $i^{th}$ client |
| $f_i$ | local loss function |
| $F_i$ | local objective function |
| $F$ | global objective function |
| $\mathbf{E}.$ | expectation on $\cdot$ |
| $\mathbf{H}$ | entropy |
| $\mathbf{P}$ | probability measure |
| $\mathbf{P}_{ef}$ | probability in exponential family |
| $\mathbf{P}_{sef}$ | probability in scaled exponential family |
| $\Omega.$ | complete set of $\cdot$ |
| $\mathcal{F}$ | generic $\sigma$-algebra |
| $\sigma(\cdot)$ | $\sigma$-algebra derived from $\cdot$ |
| $\{\Omega, \mathcal{F}\}$ | measurable space |
| $\{\Omega, \sigma(\Omega), \mathbf{P}\}$ | probability measurable space |
| $\hat{\mathbf{P}}$ | estimated probability |
| $x_i, y_i, d_i$ | input data, label data, the pairs of them |
| $w$ | global model parameter |
| $\Theta_i$ | local information |
| $\theta_i$ | personalized parameters |
| $w_{init}, \theta_{init}$ | function to initialize parameters |
| $\mu_i$ | the function to generate mean parameter |
| $s_i$ | the function to generate natural parameter |
| $x, y, \hat{w}, \theta, \mu, s$ | generic point notations |
| T,N,R,S | number of total global epochs, clients, local epochs, number of sampling clients |
| t,r | global epochs, local epochs |
| $\beta, \eta, \eta_\alpha, \lambda, \hat{\lambda}$ | scalar notations |
| $g, h, h_\lambda$ | generic function notations |
| $\mathcal{D}_g$ | Bregman divergence derived from g |
| $\mathcal{D}\mathbf{prox}$ | Bregman divergence proximal mapping |
| $\mathcal{D}\mathbf{env}$ | Bregman Moreau envelope |
| $\nabla, \mathbf{D}, \nabla^2$ | gradient, Jocobian and Hessian operator |
| $\Delta$ | deviation from mean |
| $\cdot^*$ | the Fenchel conjugate of $\cdot$ |
| $\mathcal{L}$ | averaged local test loss |
| $\mathcal{G}$ | averaged global test loss |
| $\bar{\cdot}$ | mean of $\cdot$ over clients |
| $I, I_m$ | identity mapping, identity matrix |

## A.2 Bregman Divergence

Bregman divergence is a general distance satisfying that its first-order moment estimation is the point that minimizes the expectation of the distance to all points for all measurable functions on $\mathbf{R}^d$. In other words, the given distance $\mathcal{D}$ satisfies Condition ( 15):

$$\forall X \in \{\mathbf{R}^d, \mathcal{F}, \mathbf{P}\}, \mathbf{E}[X] = \arg\min_y \mathbf{E}[\mathcal{D}(X, y)] \tag{15}$$

Eq. (16) is the definition of Bregman divergence:

$$\begin{aligned}
\mathcal{D}_g(x, y) &:= g(x) - g(y) - \langle \nabla g(y), x - y \rangle \\
&= \int_y^x \nabla g(t) - \nabla g(y) dt
\end{aligned} \tag{16}$$

where $g$ is a convex function. For convenience, in this paper, $g$ is assumed to be strictly convex, proper and differentiable, so that the equation above Eq. (16) are well-defined. In the perspective of Taylor expansion, Bregman divergence is the first-order residual of $g$ expanded at point $y$ valued at point $x$, which is the natural connection between Bregman divergence and Legendre transformation. The Bregman divergence does not satisfy the distance axiom, but it provides some of the properties we need, such as non-negative distance. Hence, the selected function $g$ should be convex. Furthermore, if one wants the distance to have a good property that $x = y \leftrightarrow \mathcal{D}_g(x, y) = 0$, one needs $g$ to be strictly convex.

## A.3 Non-Maximum Entropy

Besides, the non-maximum entropy rule approach is also worth considering, but we focus on maximum entropy prior in this section. See [63, 24, 38, 23] for additional information of non-maximum entropy assumptions.

## A.4 Future PFL

Besides the FTML, Bayesian learning, EM, and transfer learning mentioned in the main paper, neural-collapse-motivated methods and life-long learning are also promising methods to handle PFL problem [47, 30, 73].

PFL could also fucos on personalizing other characteristics about FL system, e.g., communications, resource-constrained device. For example, this paper [80] gives a data distillation (compression) [78, 70] method to reduce communication cost, and the compressed data itself contains personalize posterior information.

## A.5 Personalized Prior and MAML

Based on previous derivations, to obtain a deployable algorithm, our remaining task is to determine $\Phi$. In this section, inspired by MAML, we briefly introduce a meta-step-based implementation method. The mean parameter is used to represent the prior under SX-family prior assumption given any $\lambda$ and $g$ in this paper. The mean of the SX-family prior in Eq. (8) is used in regular term, which can be personalized in each client $i$ as $\mu_i$, corresponding to $\mu_\Phi$ in Eq. (11), as shown in Figure 1. Motivated by this, we use MAML to learn the personalized regularization (or personalized prior in Bayesian learning) in Section 5. For example, $\mathbf{meg}$ in Eq. (14) uses MAML on the Bregman-Moreau envelope $\mathcal{D}\mathbf{env}_{g^*, \lambda^{-1}} f_i$ by substituting it into $J$ in Section 2 and $\Phi$ in Eq. (11).

## A.6 Sampling Method in Bayesian Learning

Bayesian methods are a elegant solution to the complex issue of heterogeneous data, as they operate on a principle whereby the model allocates increasing attention to local data as available, and derives insight from prior information when local information is scarce. Furthermore, Bayesian modeling brings fresh probabilistic insights to PFL regularization techniques, while simultaneously providing a flexible framework for exploring novel strategies. Bayesian modeling, as well as the expectation maximuzatioin and maximum a posteriori estimate (EM-MAP) [15], provide our personalized prior approach with straightforward theoretical support, as well as more general perspectives for analysis.

Table 6: Complexity Comparison

| Complexity/Methods | FedEM | FedAvg | pFedMe | Per-FedAvg | FedAMP | pFedBreD (ours) |
|---|---|---|---|---|---|---|
| Sys. Memory | $\mathbf{O}(NM)$ | $\mathbf{O}(N)$ | $\mathbf{O}(N)$ | $\mathbf{O}(N)$ | $\mathbf{O}(N)$ | $\mathbf{O}(N)$ |
| Sys. Time | $\mathbf{O}(NTRM)$ | $\mathbf{O}(NTR)$ | $\mathbf{O}(NTRK)$ | $\mathbf{O}(NTR)$ | $\mathbf{O}(NTRK)$ | $\mathbf{O}(NTRK)$ |

Meanwhile, it addresses the cost of additional sampling in the classic and approximate Bayesian learning paradigm with MAP, the regularization method.

In Bayesian modeling, the EP global loss provides more information that we want to use for local training due to its zero-avoiding property. [56]

The sampling methods used to calculate the solution of Bayesian Model mentioned in this paper can be importance sampling, MCMC or others. In this work, we use the approximation Bayesian methods. See more details in [2]. The local training process based on regular terms differs from Bayesian learning based on sampling, *i.e.*, each time a model needs to be obtained by sampling the model distribution under the current parameters. We choose to use Bayesian MAP as a point estimation as our estimation method, thus eliminating steps such as sampling and reparameterization to improve inference efficiency. The personalized model sampled from local training can be seen as the results from random data sampling using SGD or the mean parameter directly.

## A.7  First-order Methods

There are three parts in Eq. (13) we need to deal with, and the first-order methods are as shown below:

**Jacobian Matrix of Mean:** specifically, utilizing the prior selection strategy discussed in Section 4, we have $\mathbf{D}\mu_i(w) = I - \eta\nabla^2\Phi(w)$. Using different $\Phi$ functions yields varying results. For instance, with first-order methods and the last term removed, we get the approximation $\mathbf{D}\mu_i \leftarrow I$.

**Hessian Matrix:** with first-order methods, we let $\nabla^2 g^*(\cdot) = I_m$. It happens when assuming $\theta_i$ obeys the spherical Gaussian by letting $g = \frac{1}{2}||\cdot||^2$. Moreover, we can assume $\theta_i$ obeys the general multivariable Gaussian by letting $g = \langle\cdot, \Sigma^{-1}\cdot\rangle$ and $\nabla^2 g(\cdot) = \Sigma^{-1} \succeq 0$.

**Proximal Mapping:** given $\mu_i(w)$, the proximal mapping part $\mathcal{D}\mathbf{prox}_{g^*, \lambda^{-1}}f_i(\mu_i(w))$ can be approximately solved with numerical methods, *e.g.*, gradient descent methods. In other words, we can alternately calculate $\mu_i(w)$ on each client and then fix $\mu_i(w)$ in each local epoch with EM.

## A.8  Complexity

Since the general process of our implementations, FedAMP and pFedMe are the same as shown in pFedBreD framework, these methods share the same complexity of memory/calculation, $O(N)/O(NTRK)$ as shown in Table 6. The complexities of both FedAvg and Per-FedAvg are $O(N)/O(NTR)$ since the original methods of them do not need a approximate proximal mapping solution, and therefore are free on $K$, the number of iterations to calculate the solution. The complexity of FedEM is $O(NM)/O(NTRM)$, where $M$ is the components of the distributions we assume, due to the calculation of $M$ components in each global epoch.

## A.9  Broader Impacts

In recent years, PFL has found use not only in predictive tasks like mobile device input methods but also in areas where privacy is paramount, such as healthcare and finance. However, before its widespread deployment, several critical factors must be taken into consideration.

One of the primary concerns regarding PFL is its deployment cost. It involves significant computational resources, making it a costly affair. Additionally, client transparency is an important issue that needs attention. Clients have the right to know what data is being collected and how it is used.

Another factor that complicates PFL's deployment is the differences in user behavior and hardware and software configurations between clients. These differences can affect the performance of the algorithm and require bespoke solutions for each client.

In addition, PFL's robustness is another essential aspect to consider. Real-world environments are often unpredictable and can interfere with the algorithm's performance, leading to erroneous results. Therefore, it is necessary to ensure that the algorithm is sufficiently robust before deploying it.

Lastly, even though PFL offers significant benefits, potential drawbacks should not be overlooked. All stakeholders involved in its deployment need to approach this technology with caution and forethought. By considering these factors, we can harness the power of PFL while minimizing its limitations and risks.

## B  Details of Equations

### B.1  Hidden Information

From the definition of KL divergence, we have

$$
\begin{aligned}
&\arg\min_w \mathbf{E}_i \mathbf{E}_{d_i} \mathbf{KL}(\mathbf{P}(y_i|x_i)||\hat{\mathbf{P}}(y_i|x_i, w)) \\
&= \arg\min_w \mathbf{E} \log \mathbf{P}(y_i|x_i) - \log \hat{\mathbf{P}}(y_i|x_i, w)) \\
&= \arg\min_w \mathbf{E} - \log \hat{\mathbf{P}}(y_i|x_i, w)) \\
&= \arg\max_w \mathbf{E} \log \hat{\mathbf{P}}(y_i|x_i, w))
\end{aligned}
\tag{17}
$$

This is used in Eq. (5) in the main paper.

### B.2  Bregman Divergence and X-Family

We use the SX-family due to its computational advantages. While other families of distributions may be able to handle special cases, they may not be as computationally efficient.

If proper and strictly convex function $g$ is differentiable, with $g^*$ the Fenchel conjugate function of $g$, $\mathcal{D}_g(x, y)$ the Bregman divergence, $\mu$ dual point of $s$, we have:

$$
\mathcal{D}_{g^*}(\mathcal{V}, \mu) = g^*(\mathcal{V}) + g(s) - \langle \mathcal{V}, s \rangle = \mathcal{D}_g[s, \nabla g^*(\mathcal{V})]
\tag{18}
$$

From the definition of Bregman divergence , $\nabla g(s) = \mu$ and definition of $g^*$ Fenchel conjugate on convex function $g$ ,we have:

$$
\begin{aligned}
\mathcal{D}_{g^*}(\mathcal{V}, \mu) &= g^*(\mathcal{V}) - g^*(\mu) - \langle \nabla g^*(\mu), \mathcal{V} - \mu \rangle \\
&= g^*(\mathcal{V}) - g^*(\mu) - \langle s, \mathcal{V} - \mu \rangle \\
&= g^*(\mathcal{V}) - \langle s, \mathcal{V} \rangle - g^*(\mu) + \langle \mu, s \rangle \\
&= g^*(\mathcal{V}) - \langle s, \mathcal{V} \rangle + g(s)
\end{aligned}
\tag{19}
$$

Similarly, we have $\mathcal{D}_g[s, \nabla g^*(\mathcal{V})] = g^*(\mathcal{V}) - \langle s, \mathcal{V} \rangle + g(s)$. This property is used in Eq. (3) and ( 4) in the main paper.

Table 7: Bregman divergence and exponential family. (note $\xi = \langle \cdot, \ln \cdot \rangle$)

| Name | Gaussian | Bernoulli | Possion | Exponential |
|------|----------|-----------|---------|-------------|
| Domain | $\mathbf{R}^d$ | $\{0, 1\}$ | $\mathbf{N}$ | $\mathbf{R}_{++}$ |
| $g(y)$ | $\frac{1}{2}||y||^2_{\Sigma^{-1}}$ | $\ln(1 + e^y)$ | $e^y$ | $-\ln(-y)$ |
| $\nabla g(y)$ | $y$ | $\frac{\exp\{y\}}{1+\exp\{y\}}$ | $e^y$ | $-y^{-1}$ |
| $g^*(x)$ | $\frac{1}{2}||x||^2_{\Sigma^{-1}}$ | $\xi(x) + \xi(1 - x)$ | $x \ln(x) - x$ | $-\ln(x) - 1$ |
| $\nabla g^*(x)$ | $x$ | $\ln(\frac{x}{1-x})$ | $\ln(x)$ | $-x^{-1}$ |
| $\mathcal{D}_{g^*}(x, y)$ | $\frac{1}{2}||x - y||^2_{\Sigma^{-1}}$ | $\ln(1 + e^{(1-2x)y})$ | $e^y + \xi(x) - x(y + 1)$ | $\frac{x}{y} - \ln\frac{x}{y} - 1$ |

Table 7 shows parts of the relationship between specific $g$ and related member in exponential family. See [6] for more about the relationships between $g$ that derives Bregman divergence $\mathcal{D}_g$ and related derived divergence (e.g., $\cdot\Sigma^{-1}\cdot$ & Mahalanobis distance, $\sum_\cdot \cdot \log \cdot$ & KL divergence / generalized I-divergence and etc.).

## B.3 Expectation Maximization

The details of Eq. (7) in the main paper is shown in Eq. (20).

$$\sum_i \log \mathbf{P}(y_i|x_i, w) = \sum_i \log \int \mathbf{P}(y_i, \Theta_i|x_i, w)d\Theta_i = \sum_i \int \mathbf{Q}(\Theta_i) \log \frac{\mathbf{P}(y_i, \Theta_i|x_i, w)}{\mathbf{Q}(\Theta_i)} d\Theta_i$$

$$\geq \sum_i \int \log \mathbf{Q}(\Theta_i) \frac{\mathbf{P}(y_i, \Theta_i|x_i, w)}{\mathbf{Q}(\Theta_i)} d\Theta_i = \sum_i \mathbf{E}_{\mathbf{Q}(\Theta_i)} \log \frac{\mathbf{P}(y_i, \Theta_i|x_i, w)}{\mathbf{Q}(\Theta_i)}$$

$$= \sum_i \mathbf{E}_{\mathbf{Q}(\Theta_i)} \log \mathbf{P}(y_i, \Theta_i|x_i, w) - \log \mathbf{Q}(\Theta_i)$$

$$\geq \sum_i \mathbf{E}_{\mathbf{Q}(\Theta_i)} \log \mathbf{P}(y_i, \Theta_i|x_i, w) = \sum_i \mathbf{E}_{\mathbf{Q}(\Theta_i)} [\log \hat{\mathbf{P}}(y_i|x_i, \Theta_i, w) + \log \mathbf{P}(\Theta_i|x_i, w)]$$

$$= \sum_i \mathbf{E}_{\mathbf{Q}(\Theta_i)} [\log \hat{\mathbf{P}}(y_i|x_i, \Theta_i, w) + \log \int_{y_i} \mathbf{P}(\Theta_i|d_i, w) \mathbf{P}(y_i|x_i, w)]$$

$$\geq \sum_i \mathbf{E}_{\mathbf{Q}(\Theta_i)} [\log \hat{\mathbf{P}}(y_i|x_i, \Theta_i, w) + \mathbf{E}_{y_i|x_i, w} \log \mathbf{P}(\Theta_i|d_i, w)]$$

$$(20)$$

In Eq. (20), we use the concavity of logarithmic function for the first inequality and entropy $\mathbf{H}(\mathbf{Q}(\Theta_i)) = \mathbf{E}_{\mathbf{Q}(\Theta_i)} - \log \mathbf{Q}(\Theta_i) \geq 0$ the for the second. (probability $\mathbf{Q}(\Theta_i) \in [0, 1]$; The first equal sign holds, when $\mathbf{Q}(\Theta_i) = \mathbf{P}(\Theta_i|d_i, w)$.) The last inequality is derived from the concavity of the logarithmic function.

**Why is a-posteriori distribution a prior in this modeling and problem formulation?** We assume $\Theta_i|d_i, w \sim \hat{\mathbf{P}}_{sef}(\Theta_i; \lambda, s_i(w; d_i), g)$, and have:

$$\mathbf{E}_{\mathbf{Q}(\Theta_i)} [\log \hat{\mathbf{P}}(y_i|x_i, \Theta_i, w) + \mathbf{E}_{y_i|x_i, w} \log \mathbf{P}(\Theta_i|d_i, w)]$$

$$= \mathbf{E}_{\mathbf{Q}(\Theta_i)} \log \hat{\mathbf{P}}(y_i|x_i, \Theta_i, w)$$

$$+ \mathbf{E}_{\mathbf{Q}(\Theta_i)} \mathbf{E}_{y_i|x_i, w} [\log \mathbf{P}(\Theta_i|x_i, w) + \log \mathbf{P}(y_i|\Theta_i, x_i, w) - \log \mathbf{P}(y_i|x_i, w)]$$

$$(21)$$

Optimization local problem taken on both side in any $\mathbf{Q}$ sampling, we have:

$$\arg \min_{\Theta_i} \{\log \hat{\mathbf{P}}(y_i|x_i, \Theta_i, w) + \mathbf{E}_{y_i|x_i, w} \log \mathbf{P}(\Theta_i|d_i, w)\}$$

$$= \arg \min_{\Theta_i} \{\log \hat{\mathbf{P}}(y_i|x_i, \Theta_i, w)$$

$$+ \mathbf{E}_{y_i|x_i, w} [\log \mathbf{P}(\Theta_i|x_i, w) + \log \mathbf{P}(y_i|\Theta_i, x_i, w) - \log \mathbf{P}(y_i|x_i, w)]\}$$

$$= \arg \min_{\Theta_i} \{\underbrace{\log \hat{\mathbf{P}}(y_i|x_i, \Theta_i, w)}_{\text{Predicted Likelihood}} + \mathbf{E}_{y_i|x_i, w} [\underbrace{\log \mathbf{P}(\Theta_i|x_i, w)}_{\text{Prior Distribution}} + \underbrace{\log \mathbf{P}(y_i|\Theta_i, x_i, w)}_{\text{Assumed Likelihood}}]\}$$

$$(22)$$

Thus, we do maximum a-posteriori estimation alongside added predicted likelihood, which is virtually doing assumptions on prior distribution and take mixed likelihood. Moreover, taking assumption on a-posteriori distribution leads calculation efficiency. Note that the hyperparameters should be carefully discussed.

**Bi-level optimization trick:**

$$\max_{x,y} f(x, y) \geq \max_x \max_y f(x, y)$$

$$\sum_i a_i \max f(x, y_i) = \max \sum_i a_i f(x, y_i)$$

$$(23)$$

In Eq. (8), we use the two properties of max shown in Eq. (23). Moreover, these properties are also used to build the upper bound of Eq. (10) as Eq. (11).

## B.4 Notations of Deviations

The notations are shown as follows:

$\mathcal{L}_{i,c}$: The averaged local test loss of the $i^{th}$ personalized model over its own local test with label $c$. The value equals zero on the clients without $c$-labeled data.

$\bar{\mathcal{L}}_c$: The mean of the averaged local test loss over all personalized models. Each $\mathcal{L}_{i,c}$ is weighted by the ratio of the number of own test data with label $c$.

$\mathcal{G}_{i,c}$: The averaged global test loss of the $i^{th}$ personalized model over the global test with label $c$.

$\bar{\mathcal{G}}_c$: The mean of the averaged global test loss over all personalized models.

The deviations of the averaged global and local test loss of the $i^{th}$ personalized model on class $c$: $\Delta\mathcal{G}_{i,c} = \mathcal{G}_{i,c} - \bar{\mathcal{G}}_c$ and $\Delta\mathcal{L}_{i,c} = \mathcal{L}_{i,c} - \bar{\mathcal{L}}_c$.

# C  More About Experiments

The access of all data and code is available [9] .

## C.1  More about implementations

The three implementations of $\mu_i$, *i.e.* **lg**, **meg** and **mh**, represent *loss gradient*, *memorized envelope gradient* and *memorized hybrid* respectively. *Memorized* means that we choose the gradient of Bregman-Moreau envelope $\nabla F_i(w_{i,r-1}^{(t)})$ as $\eta[w_{i,R}^{(t-1)} - \theta_{i,r-1}^{(t)}]$, where $\eta \geq 0$ is a step-size-like hyper-parameter. Each local client memorizes their own local part of the latest global model $w^{(t)}$ at the last global epochs $w_{i,R}^{(t-1)}$, instead of $w_{i,r-1}^{(t)}$ in practice.

## C.2  Variant

Based on the facts, the results in Table 1 shows the instability of our personalized models. Here we propose a variant of **mh**, shown in Eq. (24), trying to improve the robustness of personalized model on the original **mh**, which use $\Phi_i \leftarrow f_i + F_i$.

$$
\begin{aligned}
\Phi_i &\leftarrow \tilde{F}_{i,\tilde{\eta}_\alpha,\tilde{\eta}} := \tilde{\eta}_\alpha f_i \circ (\cdot - \tilde{\eta}\nabla f_i) + F_i \\
\mu_{i,r} &\leftarrow w_{i,r-1}^{(t)} - \eta\nabla\tilde{F}_{i,\tilde{\eta}_\alpha,\tilde{\eta}}(w_{i,r-1}^{(t)}) \\
&= w_{i,r-1}^{(t)} - \eta\{\tilde{\eta}_\alpha\nabla f_i[w_{i,r-1}^{(t)} - \tilde{\eta}\nabla f_i(w_{i,r-1}^{(t)})]\} - \eta\{w_{i,R}^{(t-1)} - \theta_{i,r-1}^{(t)}\}
\end{aligned}
\tag{24}
$$

This method in Eq. (24) performance almost the same as the orginal **mh** when $\eta_\alpha$ is small, but it provides flexibility to tune the hyper-parameter and decide whether to focus more on the current gradient step or the meta-gradient step by tuning $\tilde{\eta}_\alpha$ and $\tilde{\eta}$. $\tilde{\eta}_\alpha \leftarrow \eta_\alpha/\eta$ and $\tilde{\eta} \leftarrow \eta_\alpha$ are used in practice.

## C.3  Implementations of Per-FedAvg

We implement Per-FedAvg with the first-order method [20] and fine-tune the personalized model twice, with each learning step of the global and personalized step sizes.

## C.4  Details of Tricks, Datasets and Models

Tricks are shown as follows:

**FT:** fine-tuning single personalized model one more step for local test.

**AM:** aggregate momentum, the same trick used in $12^{th}$ line of Algorithm 1.(To compare more fairly between methods with single global model; $\beta = 2$ for methods and employing AM)

Datasets settings are shown as follows:

**CIFAR-10:** the whole dataset is separated into 20 clients, and each client has data of 3 classes of label. [18, 39]

---

[9] https://github.com/BDeMo/pFedBreD_public

**FEMNIST:** we use non-i.i.d. FEMNIST from LEAF benchmark with fraction of data to sample of 5% and fraction of training data of 90%. [11]

**FMNIST:** the whole fashion-MNIST dataset is separated into 100 clients, and each client has data of 2 classes of label. [65, 67]

**MNIST:** the whole MNIST dataset is separated into 100 clients, and each client has data of 3 classes of label. [65, 42]

**Sent140:** we use non-i.i.d text dataset Sent140 from LEAF benchmark with fraction of data to sample of 5%, fraction of training data of 90% and minimum number of samples per user of 3. Then we re-separate Sent140 into 10 clients with at least 10 samples. [11]

Model settings are shown as follows:

**CNN:** for the image data, we use convolutional neural network of CifarNet [28].

**DNN:** the non-linear model is 2 layers deep neural network with 100-dimension hidden layer and activation of leaky ReLU [51] and output of softmax.

**MCLR:** the linear model, multi-class linear regression, is 1 layer of linear mapping with bias, and then output with softmax.

**LSTM:** text data model consists of 2 LSTM layers [27] as feature extraction layer of 50-dimension embeding and hidden layer and 2 layers deep neural network as classifier with 100-dimension of hidden layer.

## C.5 Non-I.I.D Distribution

Figure 4 shows the non-i.i.d. distribution of MNIST, CIFAR-10, FMNIST, FEMNIST and Sent140. Sent140 is a bi-level classification so each client has two class of label data and we directly use the LEAF benchmark [11] and Dirichlet distribution of $\alpha = 0.5$ to separate users into 10 groups (See the code for more details).

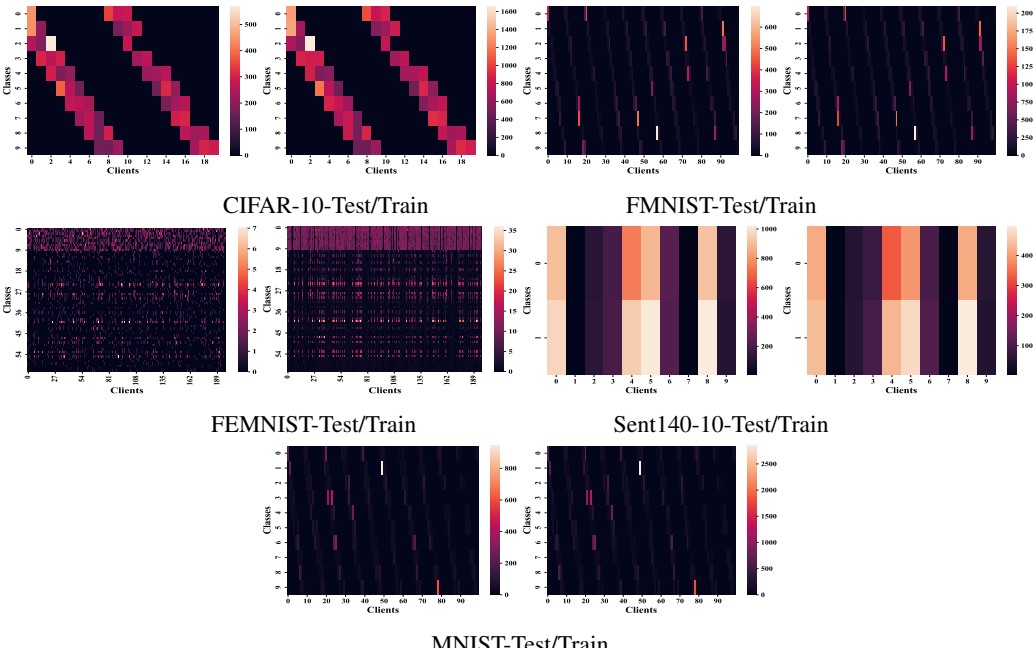

Figure 4: The visualization of the non-i.i.d. data distributions of MNIST, CIFAR-10, FMNIST, FEMNIST and Sent140.

## C.6 More About Hyper-Parameter Effect

We post the hyper-parameter effects of $\eta$ and $\lambda$ on FEMNIST, FMNIST, MNIST and Sent140 and of $\eta$ on CIFAR-10 in Figure 5- 8. We haven't put the effects of $\lambda$ on CIFAR-10 for better visualization of the effects of more sensitive eta, as well as our equipment limitations, and the fact that other non-linear models for image classification are already demonstrated on FEMNIST, FMNIST and MNIST. The results of these figures are in the same hyper-parameter settings as mentioned in Section 6.1 except the varying hyper-parameters.

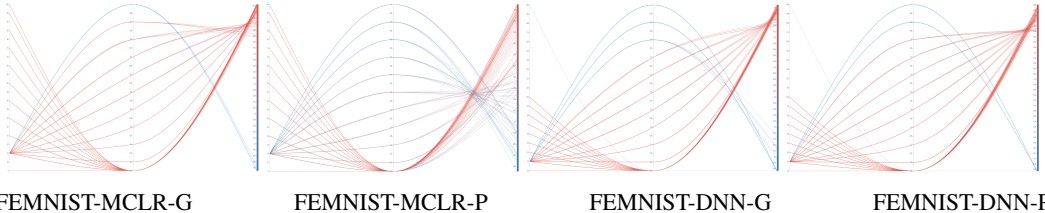

FEMNIST-MCLR-G      FEMNIST-MCLR-P      FEMNIST-DNN-G      FEMNIST-DNN-P

Figure 5: Hyper-parameter effect: The left, middle and right bars in each figure respectively represent $\lambda$, $\eta$ and test accuracy, ranges of which are respectively [0,100], [0,1] and [0,1] increasing from bottom to top (color from blue to red refers to the accuracy from 0 to 1).

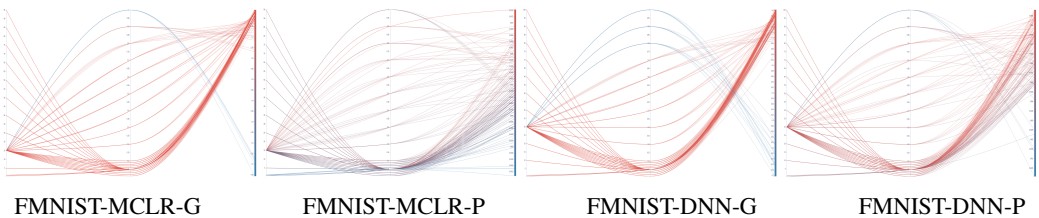

FMNIST-MCLR-G      FMNIST-MCLR-P      FMNIST-DNN-G      FMNIST-DNN-P

Figure 6: The left, middle and right bars in each figure respectively represent $\lambda$, $\eta$ and test accuracy, ranges of which are respectively [0,100], [0,1] and [0,1] increasing from bottom to top (color from blue to red).

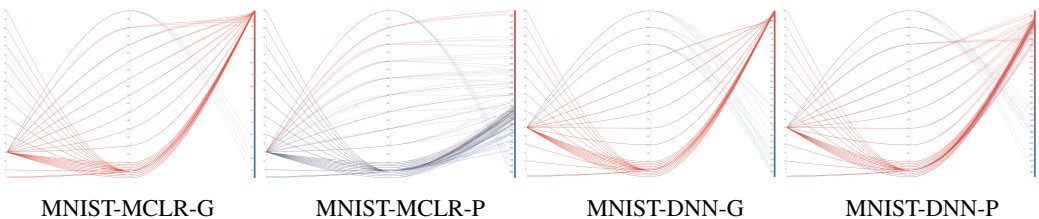

MNIST-MCLR-G      MNIST-MCLR-P      MNIST-DNN-G      MNIST-DNN-P

Figure 7: The left, middle and right bars in each figure respectively represent $\lambda$, $\eta$ and test accuracy, ranges of which are respectively [0,100], [0,1] and [0,1] increasing from bottom to top (color from blue to red).

## C.7 More about Deviation Analysis

The deviations of the global and local test on each settings are shown in Figure 9 mentioned in Section 6.2 in the main paper.

## C.8 Experiments about Instability and Robustness on Aggregation Noise and Data Heterogeneity

In this section, we experimentally demonstrate the instability of the global model in **mh** at small aggregation ratios by comparing the performances of clients with different aggregation numbers. Additionally, we also conduct experiments on different data heterogeneity settings.

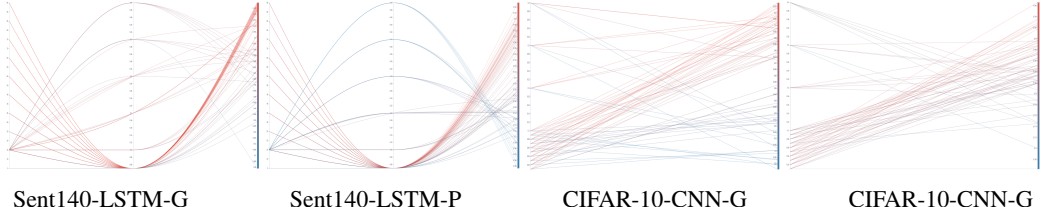

| Sent140-LSTM-G | Sent140-LSTM-P | CIFAR-10-CNN-G | CIFAR-10-CNN-G |

Figure 8: The left, middle and right bars in each figure respectively represent $\lambda$ and test accuracy, ranges of which are respectively [0,100] and [0,1] increasing from bottom to top (color from blue to red). The ranges of $\eta$ are respectively [0,0.5] and [0,0.4] in settings of CIFAR-10-CNN and Sent140.

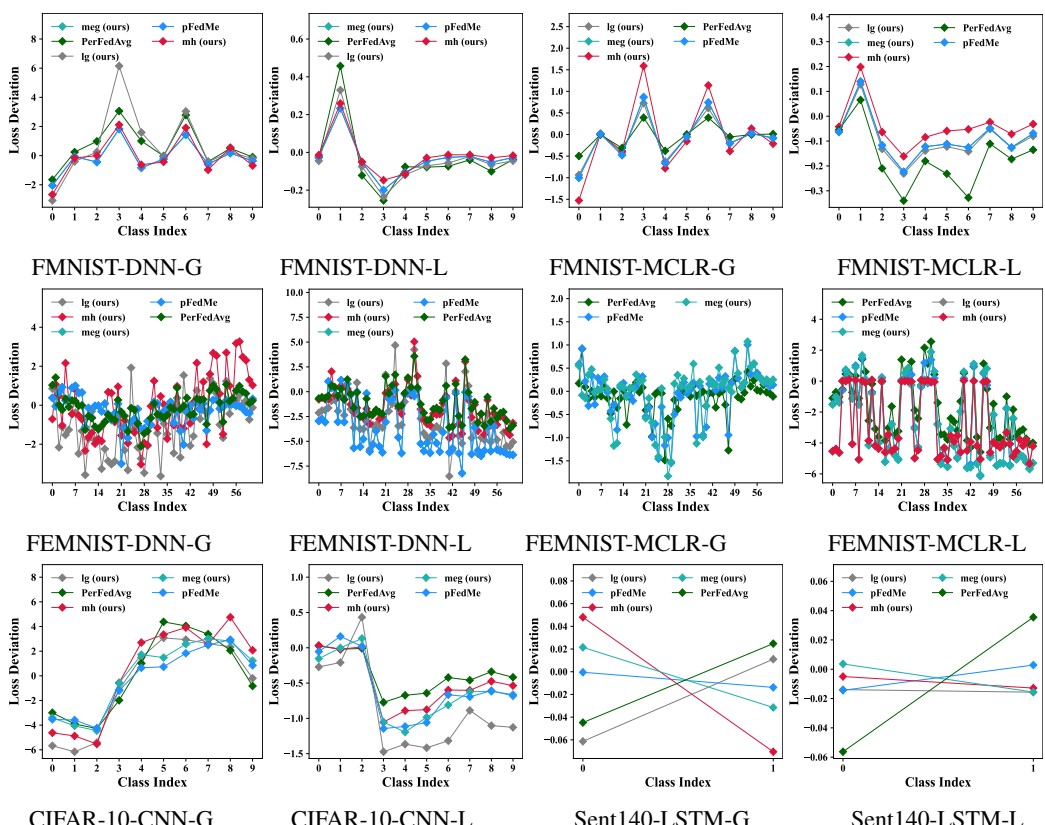

Figure 9: The loss deviation of our experiments in Section 6 on the first client on settings: FEMNIST-DNN/MCLR, FMNIST-DNN/MCLR, CIFAR-10-CNN and Sent140-LSTM.

The experimental settings in Section 6.1 of the main paper have been utilized, with the exception of the client count for aggregation at the culmination of each global epoch. To ensure clarity, we present Table 3 without well-tuning hyper-parameters (which are random selected in a narrow range with Gaussian variance of 0.01). Notably, supplementary experiments have been repeated 5 times to enhance the robustness of our analysis.

The results of experiments about different Non-IID settings are shown in Table 4. The FMNIST in these experiments are equal number of total local data with different local data distribution the distribution are shown in Figure 10. All experiments employ full aggregation of 40 clients and only 1 local epoch to get rid of the effects from aggregation noise and client drift caused by multiple local update.

An interesting example is that if the local classes are only two classes in the case of an extremely unbalanced heterogeneous distribution, the underlying local test accuracy for a personalized model

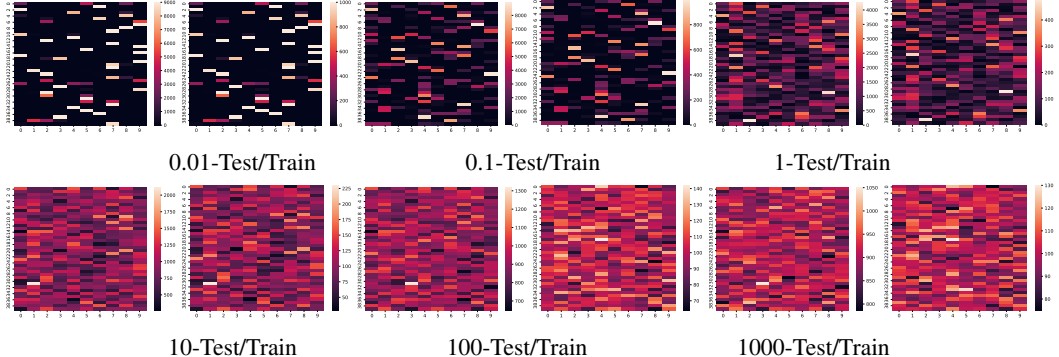

0.01-Test/Train      0.1-Test/Train      1-Test/Train

10-Test/Train      100-Test/Train      1000-Test/Train

Figure 10: Different heterogeneous distributions of FMNIST. The horizontal and vertical axes represent the different classes and clients respectively.

Table 8: Additional experiments with new baselines. (accuracy)

| Methods / Datasets & Models | FEMNIST / DNN | CIFAR-10 / CNN | Sent140 / LSTM | Average Decrease by Noise |
|---|---|---|---|---|
| Aggregation Ratio | $10\% \to 5\%$ | $20\% \to 10\%$ | $40\% \to 20\%$ | - |
| FedPAC [68] | $62.2\% \to 60.7\%$ | $78.9\% \to 77.3\%$ | $68.1\% \to 66.8\%$ | 1.5% |
| FedHN [64] | $61.1\% \to 59.6\%$ | $77.5\% \to 76.9\%$ | $71.2\% \to 70.1\%$ | 1.1% |
| Fedfomo [75] | $60.1\% \to 58.9\%$ | $71.4\% \to 70.6\%$ | $70.1\% \to 68.9\%$ | 1.1% |
| Ditto [44] | $52.9\% \to 52.2\%$ | $72.4\% \to 72.1\%$ | $71.0\% \to 70.3\%$ | 0.6% |
| mh(ours) | $\mathbf{64.9\% \to 64.3\%}$ | $\mathbf{79.4\% \to 79.1\%}$ | $\mathbf{72.0\% \to 71.8\%}$ | **0.4%** |

will be at least the probability of the maximum probability class being sampled, say 90% of the first class and 10% of the second class, then a learned knowledge model is at least 90% accurate.

### C.9  Additional Experiments

The additional experiments with more baselines are shown in Table 8 with the same settings mentioned in the Table 1.

## D  Details of Theorems

### D.1  Proof Sketch

We prove the theorems primarily through two supporting lemmas. The first lemma provides the upper bound of the global iterative error, while the second lemma restricts the upper bound of the error between the actual local update and theoretical expectation.

### D.2  Related Notations

$\cdot_{i,r}^{(t)}$ represents the $\cdot$ on $i^{th}$ client at $r^{th}$ local epoch of $t^{th}$ global epoch.

The Local Sampled Data $\tilde{d}_i \in d_i$

The Approximated Personalized Model $\tilde{\theta}_{i,r}^{(t)} := \tilde{\theta}(\mu_{i,r}^{(t)})$.

The Uniform Local Data Sampling Expectation $\mathbf{E}_{\tilde{d}_i} := \frac{1}{|d_i|}\sum_{\tilde{d}_i \in d_i}$

The Unbiased Empirical First Moment $\mathbf{E}_{\tilde{d}_i}\nabla\tilde{f}_i(\theta;\tilde{d}_i) = \nabla f_i(\theta)$

The Global Minimizer $w^*$.

The Local Minimizer $\theta_{i,r}^{*(t)} := \mathcal{D}\mathbf{prox}_{g^*,\lambda^{-1}}f_i(\mu_{i,r}^{(t)})$.

The Local Approximate Error $\Delta_{i,r}^{(t)} := \tilde{\theta}_{i,r}^{(t)} - \theta_{i,r}^{*(t)}$.

The Global Approximate Squared Error $\boldsymbol{\Delta}^{(t)} := \mathbf{E}||w^{(t)} - w^*||^2$

The Approximated Global Gradient $\mathbf{g}_{i,r}^{(t)} = \lambda \mathbf{D} \mu_{i,r}^{(t)} \nabla^2 g^*(\mu_{i,r}^{(t)})[\mu_{i,r}^{(t)} - \tilde{\theta}_{i,r}^{(t)}]$

The (first-order) Approximated Envelope Gradient: $\nabla \tilde{F}_i(w)$.

$||\cdot||_m$ is any matrix norm, with $||I||_m = \hat{u}_m$.

$\mathbb{I}_E$, indicator function on event $E$ .

The Virtual Global Gradient: $\mathbf{g}^{(t)} = \frac{1}{SR} \sum_{i \in \mathcal{S}^{(t)}} \sum_{r=1}^{R} \mathbf{g}_{i,r}^{(t)}$.

The Virtual Global Step-size: $\tilde{\alpha}_m = \alpha_m \beta R$.

The Expected Smooth [26] Coefficient of $F$ and $F_i$: $L_F,\ L_{F_i}$.

Bounded Deviation Ratio of Strategy Disturbance Coefficient $\sigma_\Phi$.

## D.3 Basic Propositions

**Proposition 1** ($\mu$-strongly convex)**.** *If $f$ is $\mu$-strongly convex, we have:*

$$\langle \nabla f(x) - \nabla f(y), x - y \rangle \geq \mu ||x - y||^2$$
$$||\nabla f(x) - \nabla f(y)|| \geq \mu ||x - y||$$

**Proposition 2** ($L$-smooth)**.** *If $f$ is $L$-smooth, we have:*

$$\langle \nabla f(x) - \nabla f(y), x - y \rangle \leq L ||x - y||^2$$
$$||\nabla f(x) - \nabla f(y)|| \leq L ||x - y||$$
$$||\nabla f(x) - \nabla f(y)||^2 \leq 2L \mathcal{D}_f(x, y)$$

**Proposition 3** (Jensen's inequality)**.** *If $f$ is convex, we have:*

$$\mathbf{E}_X f(X) \geq f(\mathbf{E}_X X)$$

*. A variant of the general one shown above:*

$$||\sum_{i=1}^{\mathcal{N}} x_i||^2 \leq \mathcal{N} \sum_{i}^{\mathcal{N}} ||x_i||^2$$

.

**Proposition 4** (triangle inequality)**.** *The triangle inequality:*

$$||A + B|| \leq ||A|| + ||B||$$

**Proposition 5** (matrix norm compatibility)**.** *The matrix norm compatibility, $A \in \mathbf{R}^{a \times b}, B \in \mathbf{R}^{b \times c}, v \in \mathbf{R}^b$:*

$$||AB||_m \leq ||A||_m ||B||_m$$
$$||Av||_m \leq ||A||_m ||v||$$

**Proposition 6** (Peter Paul inequality)**.**

$$2\langle x, y \rangle \leq \frac{1}{\epsilon} ||x||^2 + \epsilon ||y||^2$$

## D.4 General Assumptions for Analysis

**Assumption 1** (Prior selection)**.** *The given $g^*$ is $\hat{\mu}_{g^*}$-strongly convex and $\hat{L}_{g^*}$-smooth: $\hat{\mu}_{g^*} ||x - y|| \leq ||\nabla g^*(x) - \nabla g^*(y)|| < \hat{L}_{g^*} ||x - y||$. and $||\nabla^2 g^*(\cdot)||_m \leq \hat{L}_{g^*}$ (Examples are in Appendix A.7).*

**Assumption 2** (Smooth envelope assumption)**.** *For each local envelope $\mathcal{E}_i(\cdot) = [F_i \circ \mu_i^{-1}](\cdot) = \mathcal{D}\mathbf{env}_{g^*, \lambda^{-1}}(\cdot))$, we have $||\nabla \mathcal{E}_i(x) - \nabla \mathcal{E}_i(y)||^2 \leq 2\hat{L}_{\mathcal{E}_i} \mathcal{D}_{\mathcal{E}_i}(x, y)$, note that $\mathcal{E}_i$ is convex, $\mathcal{D}_{\mathcal{E}_i}(x, y) := \mathcal{E}_i(x) - \mathcal{E}_i(y) - \langle \nabla \mathcal{E}_i(y), x - y \rangle$. For simplification, we take $\hat{L}_{\mathcal{E}} := \max \hat{L}_{\mathcal{E}_i}, \forall i$ and bounded difference on optimal point $0 \leq \frac{\mathcal{D}_{\mathcal{E}_i}(\mu_i(w), \mu_i(w^*))}{\mathcal{D}_{F_i}(w, w^*)} \leq \tau$.*

**Assumption 3** (Strongly convex envelope settings). $f_i$ is $\hat{\mu}_{f_i}$-strongly convex: $\hat{\mu}_{f_i}||x - y|| \leq ||\nabla f_i(x) - \nabla f_i(y)||$, $\hat{\mu}_f = \min_i \hat{\mu}_{f_i}, \forall i$; $f_i$ is $\hat{L}_{f_i}$-smooth and non-convex : $\hat{L}_{f_i}||x - y|| \geq ||\nabla f_i(x) - \nabla f_i(y)||$,, $\hat{L}_f = \max_i \hat{L}_{f_i}, \forall i$. Therefore, we have $F_i$ is $\hat{\mu}_{F_{sc}} := \lambda \hat{\mu}_g + \hat{\mu}_f$-strongly convex or $\hat{\mu}_{F_{nc}} := \lambda \hat{\mu}_g - \hat{L}_f$-strongly convex, by tuning $\lambda$ to make $\lambda \hat{\mu}_g - \hat{L}_f > 0$. We use $\hat{\mu}_{F.}$ as the unified notation for both, for simplification.

**Assumption 4** (Bounded local error). *Since classical gradient descent is used locally, we assume a unified local error bound,* $\forall(i, r, t), ||\nabla f_i(\tilde{\theta}_{i,r}^{(t)}; d_i) + \lambda \nabla \mathcal{D}_{g^*}(\tilde{\theta}_{i,r}^{(t)}, \mu_i)|| \leq c_{i,r}^{(t)} \leq \hat{\epsilon}, \forall i$, *and a local data sampling shift variance bound* $\forall \theta, \mathbf{d} \in d_i, \mathbf{E_d}||\nabla \tilde{f}_i(\theta; \mathbf{d}) - \nabla f_i(\theta; d_i)|| \leq \gamma_{f_i} \leq \hat{\gamma}_f := \max\{\gamma_{f_i}\}, \forall i$.

**Assumption 5** (RMD meta-step function bound). $\forall i, \Phi_i$ *with limited gradient,* $||\nabla \Phi_i(\cdot)|| \leq \mathcal{G}_\Phi$, *and Hessian*$||\nabla^2 \Phi_i(w)||_m \leq \hat{\gamma}_\Phi$, *therefore,* $||\mathbf{D}\mu_i(w)||_m = ||I - \eta \nabla^2 \Phi_i(w)||_m \leq \hat{u}_m + \eta \hat{\gamma}_\Phi$.

**Assumption 6** (Bounded deviation ratio of strategy disturbance). *We assume the local training is not affected too much by the personalized prior strategies, which means we don't want a large discrepancy between the results of local strategies formulation and the calculation of local envelope gradients given the prior on each client, which may cause a significant disturbance in the local optimization objective due to the haphazard formulation of prior strategies. Given* $\forall, w, w'$, *we have:*

$$\frac{||\mathbf{D}\mu_i(w) - \mathbf{D}\mu_i(w')||_m}{||\nabla \mathcal{E}(\mu_i(w)) - \nabla \mathcal{E}(\mu_i(w'))||} \leq \sigma_\Phi \frac{\max\{||\mathbf{D}\mu_i(w)||_m, ||\mathbf{D}\mu_i(w')||_m\}}{\max\{||\nabla \mathcal{E}(\mu_i(w))||, ||\nabla \mathcal{E}(\mu_i(w'))||\}}$$

**Assumption 7** (Optimal global gradient noise bound). $||\nabla F_i(w^*)||^2 \leq \sigma_{F_i,*}^2$, *let* $\sigma_{F,*}^2 = \max_i \sigma_{F_i,*}^2, \forall i$.

**Assumption 8** (First-order approximate bound). $||\nabla F_i(w) - \nabla \tilde{F}_i(w)|| \leq \epsilon_1$

## D.5 General Lemmas

**Lemma 1** (Local Samplng Proximal Bound). *Under settings and assumptions in Section 5 and Section D.4, if $f$ is $\hat{\mu}_f$-strongly convex,* $\mathbf{E}_{\tilde{d}_i}||\Delta_{i,r}^{(t)}||^2 \leq \frac{2}{(\hat{\mu}_f + \lambda \hat{\mu}_{g^*})^2}[\frac{\hat{\gamma}_f^2}{|\tilde{d}_i|} + \hat{\epsilon}^2]$ *holds; if $f$ is $\hat{L}_f$-smooth and non-convex,* $\mathbf{E}_{\tilde{d}_i}||\Delta_{i,r}^{(t)}||^2 \leq \frac{2}{(\lambda \hat{\mu}_{g^*} - \hat{L}_f)^2}[\frac{\hat{\gamma}_f^2}{|\tilde{d}_i|} + \hat{\epsilon}^2]$ *holds, such that:*

$$\mathbf{E}_{\tilde{d}_i}||\Delta_{i,r}^{(t)}||^2 \leq \frac{2}{\hat{\mu}_{F.}^2}[\frac{\hat{\gamma}_f^2}{|\tilde{d}_i|} + \hat{\epsilon}^2]$$

*Proof.* With Proposition 1, Assumption 3 and optimal condition of $F_i(\mu_{i,r}^{(t)})$ on $\theta_{i,r}^{*(t)}$, we have:

$$||\Delta_{i,r}^{(t)}||^2 = ||\tilde{\theta}_{i,r}^{(t)} - \theta_{i,r}^{*(t)}||^2 \leq \frac{1}{\mu_{F.}^2}||\mathbf{g}_{i,r}^{(t)}||^2$$

Note that, $\mathbf{g}_{i,r}^{(t)} = \nabla \tilde{f}_i(\tilde{\theta}_{i,r}^{(t)}; \tilde{d}_i) + \lambda \nabla \mathcal{D}_{g^*}(\tilde{\theta}_{i,r}^{(t)}, \mu_{i,r}^{(t)})$. With Proposition 3 and Assumption 4, we have:

$$||\mathbf{g}_{i,r}^{(t)}||^2 = \nabla \tilde{f}_i(\tilde{\theta}_{i,r}^{(t)}; \tilde{d}_i) - \nabla f_i(\tilde{\theta}_{i,r}^{(t)}; d_i) + \nabla f_i(\tilde{\theta}_{i,r}^{(t)}; d_i) + \lambda \nabla \mathcal{D}_{g^*}(\tilde{\theta}_{i,r}^{(t)}, \mu_i)$$

$$\leq 2\{||\nabla \tilde{f}_i(\tilde{\theta}_{i,r}^{(t)}; \tilde{d}_i) - \nabla f_i(\tilde{\theta}_{i,r}^{(t)}; d_i)||^2 + ||\nabla f_i(\tilde{\theta}_{i,r}^{(t)}; d_i) + \lambda \nabla \mathcal{D}_{g^*}(\tilde{\theta}_{i,r}^{(t)}, \mu_i)||^2\}$$

$$\leq 2\{||\nabla \tilde{f}_i(\tilde{\theta}_{i,r}^{(t)}; \tilde{d}_i) - \nabla f_i(\tilde{\theta}_{i,r}^{(t)}; d_i)||^2 + \hat{\epsilon}^2\}$$

Taking expectation on both sides, combining both inequalities above, we have:

$$\mathbf{E}_{\tilde{d}_i}||\Delta_{i,r}^{(t)}||^2 \leq 2\{\frac{1}{|\tilde{d}_i|^2}\mathbf{E_d}||\sum_{\mathbf{d} \in \tilde{d}_i} \nabla \tilde{f}_i(\tilde{\theta}_{i,r}^{(t)}; \mathbf{d}) - \nabla f_i(\tilde{\theta}_{i,r}^{(t)}; d_i)||^2 + \hat{\epsilon}^2\}$$

$$\leq 2\{\frac{1}{|\tilde{d}_i|^2}\sum_{\mathbf{d} \in \tilde{d}_i} \mathbf{E_d}||\nabla \tilde{f}_i(\tilde{\theta}_{i,r}^{(t)}; \mathbf{d}) - \nabla f_i(\tilde{\theta}_{i,r}^{(t)}; d_i)||^2 + \hat{\epsilon}^2\}$$

$$\leq 2[\frac{\hat{\gamma}_f^2}{|\tilde{d}_i|} + \hat{\epsilon}^2]$$

$\square$

**Lemma 2** (Expected-Smooth Personalized Local Object). *Under settings and assumptions in Section 5 and Section D.4, the personalized local objective function is expected-smooth, such that:*

$$||\nabla F_i(w) - \nabla F_i(w')|| \leq (1 + \sigma_\Phi)(\hat{u}_m + \eta\hat{\gamma}_\Phi)||\nabla\mathcal{E}_i(\mu_i(w)) - \nabla\mathcal{E}_i(\mu_i(w'))||, \forall w, w';$$

$$||\nabla F_i(w) - \nabla F_i(w^*)||^2 \leq 2(1 + \sigma_\Phi)^2(\hat{u}_m + \eta\hat{\gamma}_\Phi)^2\hat{L}_{\mathcal{E}_i}\mathcal{D}_\mathcal{E}(\mu_i(w), \mu_i(w^*)) \leq 2\hat{L}_{F_i}\mathcal{D}_{F_i}(w, w^*);$$

$$\mathbf{E}_i||\nabla F_i(w) - \nabla F_i(w^*)||^2 \leq 2\hat{L}_F[F(w) - F(w^*)] = 2\hat{L}_F\mathcal{D}_F(w, w^*),$$

*where $\hat{L}_{F_i} := \tau(1 + \sigma_\Phi)^2(\hat{u}_m + \eta\hat{\gamma}_\Phi)^2\hat{L}_{\mathcal{E}_i}$ and $\hat{L}_F = \max \hat{L}_{F_i}, \forall i$.*

*Proof.* With Assumption 5 and Assumption 6, we have:

$$||\nabla F_i(w) - \nabla F_i(w')||^2 = ||\mathbf{D}\mu_i(w)\nabla\mathcal{E}(\mu_i(w)) - \mathbf{D}\mu_i(w')\nabla\mathcal{E}(\mu_i(w'))||$$
$$=||\mathbf{D}\mu_i(w)\nabla\mathcal{E}(\mu_i(w)) - \mathbf{D}\mu_i(w')\nabla\mathcal{E}(\mu_i(w')) + \mathbf{D}\mu_i(w)\nabla\mathcal{E}(\mu_i(w')) - \mathbf{D}\mu_i(w)\nabla\mathcal{E}(\mu_i(w'))||$$
$$\leq||\mathbf{D}\mu_i(w)[\nabla\mathcal{E}(\mu_i(w)) - \nabla\mathcal{E}(\mu_i(w'))]|| + ||[\mathbf{D}\mu_i(w') - \mathbf{D}\mu_i(w)]\nabla\mathcal{E}(\mu_i(w'))||$$
$$\leq||\mathbf{D}\mu_i(w)||_m||[\nabla\mathcal{E}(\mu_i(w)) - \nabla\mathcal{E}(\mu_i(w'))]|| + ||[\mathbf{D}\mu_i(w') - \mathbf{D}\mu_i(w)]||_m||\nabla\mathcal{E}(\mu_i(w'))||$$
$$\leq \max\{||\mathbf{D}\mu_i(w)||, ||\mathbf{D}\mu_i(w')||\}||\nabla\mathcal{E}(\mu_i(w)) - \nabla\mathcal{E}(\mu_i(w'))||$$
$$+ \max\{||\nabla\mathcal{E}(\mu_i(w))||, ||\nabla\mathcal{E}(\mu_i(w'))||\}|[\mathbf{D}\mu_i(w') - \mathbf{D}\mu_i(w)]||_m$$
$$\leq \max\{||\mathbf{D}\mu_i(w)||, ||\mathbf{D}\mu_i(w')||\}||\nabla\mathcal{E}(\mu_i(w)) - \nabla\mathcal{E}(\mu_i(w'))||$$
$$+ \sigma_\Phi \max\{||\mathbf{D}\mu_i(w)||_m, ||\mathbf{D}\mu_i(w')||_m\}||\nabla\mathcal{E}(\mu_i(w)) - \nabla\mathcal{E}(\mu_i(w'))||$$
$$\leq(1 + \sigma_\Phi)(\hat{u}_m + \eta\hat{\gamma}_\Phi)||\nabla\mathcal{E}(\mu_i(w)) - \nabla\mathcal{E}(\mu_i(w'))||$$

where the first two inequalities is by Proposition 4 and Proposition 5.

With the first inequality in our lemma is proven. With the proven one and Assumption 2, we have:

$$||\nabla F_i(w) - \nabla F_i(w^*)||^2 \leq 2(1 + \sigma_\Phi)^2(\hat{u}_m + \eta\hat{\gamma}_\Phi)^2\hat{L}_{\mathcal{E}_i}\mathcal{D}_\mathcal{E}(\mu_i(w), \mu_i(w^*))$$
$$\leq 2(1 + \sigma_\Phi)^2(\hat{u}_m + \eta\hat{\gamma}_\Phi)^2\hat{L}_{\mathcal{E}_i}\tau\mathcal{D}_{F_i}(w, w^*)$$
$$\leq 2\hat{L}_{F_i}\mathcal{D}_{F_i}(w, w^*);$$
$$\mathbf{E}_i||\nabla F_i(w) - \nabla F_i(w^*)||^2 \leq 2\hat{L}_F[F(w) - F(w^*)] = 2\hat{L}_F\mathcal{D}_F(w, w^*)$$

where the client sampling expectation is taken in the final inequality. $\square$

**Lemma 3** (RMD Personalized Prior Bound). *Under settings and assumptions in Section 5 and Section D.4, the relationship between $||\nabla F_i(w)||$ and $||\nabla\mathcal{E}_i(\mu_i(w))||$ is:*

$$||\nabla F_i(w)|| \leq (\hat{u}_m + \eta\hat{\gamma}_\Phi)||\nabla\mathcal{E}_i(\mu_i(w))|| \leq \lambda\hat{L}_{g^*}(\hat{u}_m + \eta\hat{\gamma}_\Phi)||\mu_i(w) - \theta_i^*||$$

*Proof.* Applying Proposition 5 and Assumption 5, $\nabla F_i(w) = \mathbf{D}\mu_i(w)\nabla\mathcal{E}_i(\mu_i(w))$, it's easy to prove the first inequality. Rewriting $\nabla\mathcal{E}_i(\mu_i(w))$ in detail as shown following, applying Proposition 5 and Assumption 1, the final inequality is proven:

$$\nabla\mathcal{E}_i(\mu_i(w)) = \lambda\nabla^2 g^*(\mu_i(w))[\mu_i(w) - \theta_i^*]$$

$\square$

**Lemma 4** (Local Objective's Client Sampling Error Bound). *Under settings and assumptions in Section 5 and Section D.4, the upper bound of local sampling error is:*

$$\mathbf{E}_{\mathcal{S}^t}||\frac{1}{S}\sum_{i\in\mathcal{S}^{(t)}}\nabla F_i(w^{(t)}) - \nabla F(w^{(t)})||^2 \leq \frac{N/S - 1}{N - 1}\sum_i^N\frac{1}{N}||\nabla F_i(w^{(t)}) - \nabla F(w^{(t)})||^2$$

*, where $|\mathcal{S}^{(t)}| = S, \forall t$.*

*Proof.* This lemma is the same lemma in [46, 65].

$$\mathbf{E}_{\mathcal{S}^t}||\frac{1}{S}\sum_{i\in\mathcal{S}^{(t)}}\nabla F_i(w^{(t)}) - \nabla F(w^{(t)})||^2 = \frac{1}{S^2}\mathbf{E}_{\mathcal{S}^{(t)}}||\sum_{i\in[N]}\mathbb{I}_{i\in\mathcal{S}^{(t)}}\nabla F_i(w^{(t)}) - \nabla F(w^{(t)})||^2$$

$$=\frac{1}{S^2}[\sum_{i\in[N]}\mathbf{E}_{\mathcal{S}^{(t)}}[\mathbb{I}_{i\in\mathcal{S}^{(t)}}]||\nabla F_i(w^{(t)}) - \nabla F(w^{(t)})||^2$$

$$+\sum_{i\neq j}\mathbf{E}_{\mathcal{S}^{(t)}}[\mathbb{I}_{i\in\mathcal{S}^{(t)}},\mathbb{I}_{j\in\mathcal{S}^{(t)}}]\langle\nabla F_i(w^{(t)}) - \nabla F(w^{(t)}),\nabla F_j(w^{(t)}) - \nabla F(w^{(t)})\rangle]$$

$$=\frac{1}{SN}\sum_{i}^{N}||\nabla F_i(w^{(t)}) - \nabla F(w^{(t)})||^2$$

$$+\sum_{i\neq j}\frac{S-1}{SN(N-1)}\langle\nabla F_i(w^{(t)}) - \nabla F(w^{(t)}),\nabla F_j(w^{(t)}) - \nabla F(w^{(t)})\rangle$$

$$=\frac{1}{SN}(1-\frac{S-1}{N-1})\sum_{i\in[N]}||\nabla F_i(w^{(t)}) - \nabla F(w^{(t)})||^2$$

$$=\frac{N/S-1}{N-1}\sum_{i\in[N]}\frac{1}{N}||\nabla F_i(w^{(t)}) - \nabla F(w^{(t)})||^2$$

where $\mathbb{I}_. \in \{0,1\}$ is indicator function, $\mathbf{E}_{\mathcal{S}^{(t)}}[\mathbb{I}_{i\in\mathcal{S}^{(t)}}] = \frac{S}{N}$ and $\mathbf{E}_{\mathcal{S}^{(t)}}[\mathbb{I}_{i\in\mathcal{S}^{(t)}},\mathbb{I}_{j\in\mathcal{S}^{(t)}}] = \frac{S(S-1)}{N(N-1)}, \forall i \neq j$. Note that:

$$\sum_{i}^{N}||\nabla F_i(w^{(t)}) - \nabla F(w^{(t)})||^2 + \sum_{i\neq j}\langle\nabla F_i(w^{(t)}) - \nabla F(w^{(t)}),\nabla F_j(w^{(t)}) - \nabla F(w^{(t)})\rangle = 0.$$

$\square$

**Lemma 5** (Variance of Global Aggregation on Client Sampling Bound). *Under settings and assumptions in Section 5 and Section D.4, the upper bound of gradient aggregation variance is:*

$$\mathbf{E}_i||\nabla F_i(w) - \nabla F(w)||^2 \leq \mathbf{E}_i||\nabla F_i(w)||^2 \leq 4\hat{L}_F\mathcal{D}_F(w,w^*) + 2\sigma_{F,*}{}^2$$

*Proof.*

$$\mathbf{E}_i||\nabla F_i(w) - \nabla F(w)||^2 \leq \mathbf{E}_i||\nabla F_i(w)||^2 \leq 2\mathbf{E}_i[||\nabla F_i(w) - \nabla F_i(w^*)||^2 + ||\nabla F_i(w^*)||^2]$$

$$\leq 4\hat{L}_F\mathcal{D}_F(w,w^*) + 2\sigma_{F,*}{}^2$$

where the first inequality is by $\mathbf{E}[||X||^2] = \mathbf{E}[||X - \mathbf{E}[X]||^2] + \mathbf{E}[||X||]^2$, the second one is by Proposition 3 and the final one is by Lemma 2 and Assumption 7. $\square$

## D.6 Supporting Lemmas

**Lemma 6** (Global Iteration Bound). *Under settings and assumptions in Section 5 and Section D.4, the upper bound of global iteration error is:*

$$\mathbf{E}_{\cdot|t}||w^{(t+1)} - w^*||^2 \leq (1 - \frac{\tilde{\alpha}_m\hat{\mu}_{F.}}{2})||w^{(t)} - w^*||^2 + \frac{3\tilde{\alpha}_m^2 + 2\tilde{\alpha}_m/\hat{\mu}_{F.}}{NR}\sum_{i,r}^{N,R}||\mathbf{g}_{i,r}^{(t)} - \nabla F_i(w^{(t)})||$$

$$+ 3\tilde{\alpha}_m^2\mathbf{E}_{\cdot|t}||\frac{1}{S}\sum_{i\in\mathcal{S}^{(t)}}\nabla F_i(w^{(t)}) - \nabla F(w^{(t)})||^2$$

$$+ (6\tilde{\alpha}_m^2\hat{L}_F - 2\tilde{\alpha}_m)\mathbf{E}\mathcal{D}_F(w^{(t)},w^*)$$

*Proof.* To separate the norm, we have:

$$\mathbf{E}_{\cdot|t}||w^{(t+1)} - w^*||^2 = \mathbf{E}_{\cdot|t}[||w^{(t)} - \tilde{\alpha}_m\mathbf{g}^{(t)} - w^*||^2]$$

$$= ||w^{(t)} - w^*||^2 - 2\tilde{\alpha}_m\mathbf{E}_{\cdot|t}[\langle\mathbf{g}^{(t)},w^{(t)} - w^*\rangle] + \tilde{\alpha}_m^2\mathbf{E}_{\cdot|t}[||\mathbf{g}^{(t)}||^2]$$

The second term:

$$-2\tilde{\alpha}_m \mathbf{E}_{\cdot|t}[\langle \mathbf{g}^{((t)}, w^{(t)} - w^* \rangle] = -2\tilde{\alpha}_m \langle \mathbf{E}_{\cdot|t}\mathbf{g}^{((t)}, w^{(t)} - w^* \rangle$$

$$= -2\tilde{\alpha}_m \frac{1}{NR} \sum_{i,r}^{N,R} [\langle \mathbf{g}_{i,r}^{(t)} - \nabla F_i(w^{(t)}), w^{(t)} - w^* \rangle + \langle \nabla F_i(w^{(t)}), w^{(t)} - w^* \rangle]$$

$$= \frac{\tilde{\alpha}_m}{NR} \sum_{i,r}^{N,R} [-2\langle \mathbf{g}_{i,r}^{(t)} - \nabla F_i(w^{(t)}), w^{(t)} - w^* \rangle] - 2\tilde{\alpha}_m \mathbf{E}_i \langle \nabla F_i(w^{(t)}), w^{(t)} - w^* \rangle$$

Each of the two factors of the second term is bounded (note that $\mathbf{E}_i = \frac{1}{N} \sum_{i=1}^{N}$ is discussed):

$$-\mathbf{E}_i \langle \nabla F_i(w^{(t)}), w^{(t)} - w^* \rangle \le -\mathbf{E}\mathcal{D}_F(w^{(t)}, w^*) - \mathbf{E}\frac{\hat{\mu}_{F_\cdot}}{2}||w^{(t)} - w^*||^2$$

$-2\langle \mathbf{g}_{i,r}^{(t)} - \nabla F_i(w^{(t)}), w^{(t)} - w^* \rangle \le \frac{2}{\tilde{\mu}_{F_\cdot}}||\mathbf{g}_{i,r}^{(t)} - \nabla F_i(w^{(t)})|| + \frac{\hat{\mu}_{F_\cdot}}{2}||w^{(t)} - w^*||^2$ where the first inequality is by Proposition 1 and the second one is by Proposition 6.

The third term:

$$\mathbf{E}_{\cdot|t}||\mathbf{g}^{(t)}||^2 = \mathbf{E}_{\cdot|t}||\frac{1}{SR} \sum_{i,r}^{\mathcal{S}^{(t)},R} \mathbf{g}_{i,r}^{(t)}||^2 \le 3\mathbf{E}_{\cdot|t}[||\frac{1}{SR} \sum_{i,r}^{\mathcal{S}^{(t)},R} \mathbf{g}_{i,r}^{(t)} - \nabla F_i(w^{(t)})||^2$$

$$+ ||\frac{1}{S} \sum_{i \in \mathcal{S}^{(t)}} \nabla F_i(w^{(t)}) - \nabla F(w^{(t)})||^2 + ||\nabla F(w^{(t)})||^2]$$

$$\le 3[\frac{1}{NR} \sum_{i,r}^{N,R} ||\mathbf{g}_{i,r}^{(t)} - \nabla F_i(w^{(t)})||^2$$

$$+ \mathbf{E}_{\cdot|t}||\frac{1}{S} \sum_{i \in \mathcal{S}^{(t)}} \nabla F_i(w^{(t)}) - \nabla F(w^{(t)})||^2 + 2\hat{L}_F \mathbf{E}\mathcal{D}_F(w^{(t)}, w^*)]$$

where the first inequality is by Proposition 3 the second one is by $\nabla F(w^{(t)}) = \nabla F(w^{(t)}) - \nabla F(w^*)$ and Lemma 2.

Thus, if we combine each term back into the separation at the very beginning of this proof, the lemma is proven. $\qquad\square$

**Lemma 7** (Local-Global Client Drift Bound). *Under settings and assumptions in Section 5 and Section D.4, by choosing a proper $\tilde{\alpha}_m \le \frac{\beta}{\sqrt{2\dot{c}}}$, the client drift bound is:*

$$\frac{1}{NR} \sum_{i,r}^{N,R} \mathbf{E}_{\cdot|t,i}||\mathbf{g}_{i,r}^{(t)} - \nabla F_i(w^{(t)})||^2 \le \dot{\delta} + e\dot{c}\alpha_m^2 2^{R+1} \{(1 + 2R)\mathbf{E}_{\cdot|t,i}[||\nabla F_i(w^{(t)})||^2] + \dot{\delta}\}$$

*where $\dot{\delta} = 4[\lambda \frac{\hat{L}_{g^*}}{\hat{\mu}_{F_\cdot}}(\hat{u}_m + \eta\hat{\gamma}_\Phi)]^2 (\frac{\hat{\gamma}_f^2}{|\bar{d}_i|} + \hat{\epsilon}^2) + 16[(1 + \sigma_\Phi)\hat{L}_{\mathcal{E}}(\hat{u}_m + \eta\hat{\gamma}_\Phi)\hat{\gamma}_\Phi]^2$ and $\dot{c} = 4[(1 + \sigma_\Phi)\hat{L}_{\mathcal{E}}(\hat{u}_m + \eta\hat{\gamma}_\Phi)]^2$.*

*Proof.*

$$||\mathbf{g}_{i,r}^{(t)} - \nabla F_i(w^{(t)})||^2 \leq 2[||\mathbf{g}_{i,r}^{(t)} - \nabla F_i(w_{i,r}^{(t)})||^2 + ||\nabla F_i(w_{i,r}^{(t)}) - \nabla F_i(w^{(t)})||^2]$$

$$\leq 2\{[\lambda \hat{L}_{g^*}(\hat{u}_m + \eta\hat{\gamma}_\Phi)]^2||\Delta_{i,r}^{(t)}||^2$$

$$+ (1 + \sigma_\Phi)^2(\hat{u}_m + \eta\hat{\gamma}_\Phi)^2||\nabla \mathcal{E}_i(\mu_i(w_{i,r}^{(t)})) - \nabla \mathcal{E}_i(\mu_i(w^{(t)}))||^2]\}$$

$$\leq 2\{[\lambda \hat{L}_{g^*}(\hat{u}_m + \eta\hat{\gamma}_\Phi)]^2||\Delta_{i,r}^{(t)}||^2 + [(1 + \sigma_\Phi)\hat{L}_{\mathcal{E}_i}(\hat{u}_m$$

$$+ \eta\hat{\gamma}_\Phi)]^2||\mu_i(w_{i,r}^{(t)}) - \mu_i(w^{(t)})||^2]\} \tag{25}$$

$$\leq 2\{[\lambda \hat{L}_{g^*}(\hat{u}_m + \eta\hat{\gamma}_\Phi)]^2||\Delta_{i,r}^{(t)}||^2 + [(1 + \sigma_\Phi)\hat{L}_{\mathcal{E}_i}(\hat{u}_m$$

$$+ \eta\hat{\gamma}_\Phi)]^2[2||w_{i,r}^{(t)} - w^{(t)}||^2 + 2||\nabla^2\Phi(w_{i,r}^{(t)}) - \nabla^2\Phi(w^{(t)})||^2]\}$$

$$\leq 2\{[\lambda \hat{L}_{g^*}(\hat{u}_m + \eta\hat{\gamma}_\Phi)]^2||\Delta_{i,r}^{(t)}||^2 + 2[(1 + \sigma_\Phi)\hat{L}_{\mathcal{E}_i}(\hat{u}_m + \eta\hat{\gamma}_\Phi)]^2||w_{i,r}^{(t)} - w^{(t)}||^2$$

$$+ 8[(1 + \sigma_\Phi)\hat{L}_{\mathcal{E}_i}(\hat{u}_m + \eta\hat{\gamma}_\Phi)\hat{\gamma}_\Phi]^2\}$$

where the first inequality is by Proposition 3, the second one is by Lemma 2, the third one is by Assumption 2 and Proposition 2, the fourth one is by Proposition 3 and bringing in Equation ( 11) and the final one is by Assumption 5.

With Lemma 1, we have:

$$\mathbf{E}_{\cdot|t,i}||\mathbf{g}_{i,r}^{(t)} - \nabla F_i(w^{(t)})||^2 \leq 4[\lambda\frac{\hat{L}_{g^*}}{\hat{\mu}_{F.}}(\hat{u}_m + \eta\hat{\gamma}_\Phi)]^2(\frac{\hat{\gamma}_f^2}{|\tilde{d}_i|} + \hat{\epsilon}^2)$$

$$+ 16[(1 + \sigma_\Phi)\hat{L}_{\mathcal{E}_i}(\hat{u}_m + \eta\hat{\gamma}_\Phi)\hat{\gamma}_\Phi]^2 + 4[(1 + \sigma_\Phi)\hat{L}_{\mathcal{E}_i}(\hat{u}_m + \eta\hat{\gamma}_\Phi)]^2\mathbf{E}_{\cdot|t,i}||w_{i,r}^{(t)} - w^{(t)}||^2$$

For simplification: $\mathbf{E}_{\cdot|t,i}||\mathbf{g}_{i,r}^{(t)} - \nabla F_i(w^{(t)})||^2 \leq \dot{\delta} + \dot{c}\mathbf{E}_{\cdot|t,i}||w_{i,r}^{(t)} - w^{(t)}||^2$

The second term:

$$\mathbf{E}_{\cdot|t,i}||w_{i,r}^{(t)} - w^{(t)}||^2 = \mathbf{E}_{\cdot|t,i}[||w_{i,r-1}^{(t)} - w^{(t)} - \alpha_m\mathbf{g}_{i,r-1}^{(t)}||^2]$$

$$\leq 2\mathbf{E}_{\cdot|t,i}[||w_{i,r-1}^{(t)} - w^{(t)} - \alpha_m\nabla F_i(w^{(t)})||^2$$

$$+ \alpha_m^2||\mathbf{g}_{i,r-1}^{(t)} - \nabla F_i(w^{(t)})||^2)]$$

$$\leq 2(1 + \frac{1}{2R})\mathbf{E}_{\cdot|t,i}[||w_{i,r-1}^{(t)} - w^{(t)}||^2] + 2(1 + 2R)\alpha_m^2\mathbf{E}_{\cdot|t,i}[||\nabla F_i(w^{(t)})||^2]$$

$$+ 2\alpha_m^2[\dot{\delta} + \dot{c}\mathbf{E}_{\cdot|t,i}||w_{i,r-1}^{(t)} - w^{(t)}||^2]$$

$$\leq 2(1 + \frac{1}{2R} + \alpha_m^2\dot{c})\mathbf{E}_{\cdot|t,i}[||w_{i,r-1}^{(t)} - w^{(t)}||^2]$$

$$+ 2(1 + 2R)\alpha_m^2\mathbf{E}_{\cdot|t,i}[||\nabla F_i(w^{(t)})||^2] + 2\alpha_m^2\dot{\delta}$$

$$\leq 2(1 + \frac{1}{R})\mathbf{E}_{\cdot|t,i}[||w_{i,r-1}^{(t)} - w^{(t)}||^2] + 2(1 + 2R)\alpha_m^2\mathbf{E}_{\cdot|t,i}[||\nabla F_i(w^{(t)})||^2]$$

$$+ 2\alpha_m^2\dot{\delta}$$

where the first inequality is by Proposition 3, the second one is by Proposition 6 and the simplified inequality and the final one is by choose $\tilde{\alpha}_m^2 \leq \frac{\beta^2}{2\dot{c}}$, and $\alpha_m^2\dot{c} \leq \frac{1}{2R^2} \leq \frac{1}{2R}$.

To recursively unroll:

$$\mathbf{E}_{\cdot|t,i}||w_{i,r}^{(t)} - w^{(t)}||^2$$

$$\leq \{(1 + 2R)\alpha_m^2\mathbf{E}_{\cdot|t,i}[||\nabla F_i(w^{(t)})||^2] + \alpha_m^2\dot{\delta}\}\sum_{\tilde{r}=0}^{r} 2^{\tilde{r}+1}(1 + \frac{1}{R})^{\tilde{r}}$$

$$\leq \{(1 + 2R)\alpha_m^2\mathbf{E}_{\cdot|t,i}[||\nabla F_i(w^{(t)})||^2] + \alpha_m^2\dot{\delta}\}\sum_{\tilde{r}=0}^{R-1} 2^{\tilde{r}+1}(1 + \frac{1}{R})^{\tilde{r}} \tag{26}$$

$$\leq \alpha_m^2 e2^{R+1}\{(1 + 2R)\mathbf{E}_{\cdot|t,i}[||\nabla F_i(w^{(t)})||^2] + \dot{\delta}\}$$

Thus, bringing in the recursively unrolled inequality back into the simplified one, the lemma's proven. $\qquad\square$

### D.7 Proof of Theorems

#### D.7.1 Proof of Theorem 1

The proof of Theorem 1 is shown as followings:

*Proof.* With Lemma 6, we have:

$$
\begin{aligned}
\mathbf{\Delta}^{(t+1)} := \mathbf{E}||w^{(t+1)} - w^*||^2 \\
\leq (1 - \frac{\tilde{\alpha}_m \hat{\mu}_{F.}}{2})\mathbf{\Delta}^{(t)} + \frac{3\tilde{\alpha}_m^2 + 2\tilde{\alpha}_m/\hat{\mu}_{F.}}{NR} \sum_{i,r}^{N,R} ||\mathbf{g}_{i,r}^{(t)} - \nabla F_i(w^{(t)})||^2 \\
+ 3\tilde{\alpha}_m^2 \mathbf{E}_{\cdot|t}||\frac{1}{S} \sum_{i \in \mathcal{S}^{(t)}} \nabla F_i(w^{(t)}) - \nabla F(w^{(t)})||^2 + (6\tilde{\alpha}_m^2 \hat{L}_F - 2\tilde{\alpha}_m)\mathbf{E}\mathcal{D}_F(w^{(t)}, w^*)
\end{aligned}
$$

With Lemma 4, we have:

$$
\begin{aligned}
\mathbf{\Delta}^{(t+1)} \leq (1 - \frac{\tilde{\alpha}_m \hat{\mu}_{F.}}{2})\mathbf{\Delta}^{(t)} + \frac{3\tilde{\alpha}_m^2 + 2\tilde{\alpha}_m/\hat{\mu}_{F.}}{NR} \sum_{i,r}^{N,R} ||\mathbf{g}_{i,r}^{(t)} - \nabla F_i(w^{(t)})||^2 \\
+ 3\tilde{\alpha}_m^2 \frac{N/S - 1}{N - 1}\mathbf{E}_i||\nabla F_i(w^{(t)}) - \nabla F(w^{(t)})||^2 + (6\tilde{\alpha}_m^2 \hat{L}_F - 2\tilde{\alpha}_m)\mathbf{E}\mathcal{D}_F(w^{(t)}, w^*)
\end{aligned}
$$

With Lemma 5, we have:

$$
\mathbf{E}_i||\nabla F_i(w) - \nabla F(w)||^2 \leq 4\hat{L}_F \mathcal{D}_F(w, w^*) + 2\sigma_{F,*}^2
$$

Thus, the inequality is:

$$
\begin{aligned}
\mathbf{\Delta}^{(t+1)} \leq (1 - \frac{\tilde{\alpha}_m \hat{\mu}_{F.}}{2})\mathbf{\Delta}^{(t)} + \frac{3\tilde{\alpha}_m^2 + 2\tilde{\alpha}_m/\hat{\mu}_{F.}}{NR} \sum_{i,r}^{N,R} ||\mathbf{g}_{i,r}^{(t)} - \nabla F_i(w^{(t)})||^2 \\
+ 3\tilde{\alpha}_m^2 \frac{N/S - 1}{N - 1}[4\hat{L}_F \mathbf{E}\mathcal{D}_F(w^{(t)}, w^*) + 2\sigma_{F,*}^2] + (6\tilde{\alpha}_m^2 \hat{L}_F - 2\tilde{\alpha}_m)\mathbf{E}\mathcal{D}_F(w^{(t)}, w^*)
\end{aligned}
\tag{27}
$$

With Lemma 7 and $\tilde{\alpha}_m \leq \frac{\beta}{\sqrt{2\dot{c}}}$, by taking full expectation of all variables noted by $\mathbf{E}$, we have:

$$
\begin{aligned}
\frac{1}{NR} \sum_{i,r}^{N,R} \mathbf{E}||\mathbf{g}_{i,r}^{(t)} - \nabla F_i(w^{(t)})||^2 \leq & e\dot{c}\alpha_m^2 2^{R+1}(1+2R)\mathbf{E}_i[||\nabla F_i(w^{(t)})||^2] + (e\dot{c}\alpha_m^2 2^{R+1} + 1)\dot{\delta} \\
\leq & e\dot{c}\alpha_m^2 2^{R+1}(1+2R)\mathbf{E}_i[2||\nabla F_i(w^{(t)}) - \nabla F_i(w^*))||^2 \\
& + 2||\nabla F_i(w^*))||^2] + (e\dot{c}\alpha_m^2 2^{R+1} + 1)\dot{\delta} \\
\leq & e\dot{c}\alpha_m^2 2^{R+3}(1+2R)\hat{L}_F \mathbf{E}\mathcal{D}_F(w^{(t)}, w^*) \\
& + e\dot{c}\alpha_m^2 2^{R+2}(1+2R)\sigma_{F,*}^2 + (e\dot{c}\alpha_m^2 2^{R+1} + 1)\dot{\delta}
\end{aligned}
$$

where the second inequality is by Proposition 3 and the final one is using Lemma 2 and Assumption 7. With this inequality, Equation (27) turns into:

$$
\begin{aligned}
\mathbf{\Delta}^{(t+1)} \leq & (1 - \frac{\tilde{\alpha}_m \hat{\mu}_{F.}}{2})\mathbf{\Delta}^{(t)} + 6\tilde{\alpha}_m^2 \frac{N/S - 1}{N - 1}\sigma_{F,*}^2 \\
& + (3\tilde{\alpha}_m^2 + 2\tilde{\alpha}_m/\hat{\mu}_{F.})[e\dot{c}\tilde{\alpha}_m^2 \frac{2^{R+2}(1+2R)}{\beta^2 R^2}\sigma_{F,*}^2 + (e\dot{c}\tilde{\alpha}_m^2 \frac{2^{R+1}}{\beta^2 R^2} + 1)\dot{\delta}] \\
& + \{e\dot{c}(3\tilde{\alpha}_m + 2/\hat{\mu}_{F.})\tilde{\alpha}_m^3 \frac{2^{R+3}(1+2R)}{\beta^2 R^2}\hat{L}_F + 12\tilde{\alpha}_m^2 \frac{N/S - 1}{N - 1}\hat{L}_F \\
& + 6\tilde{\alpha}_m^2 \hat{L}_F - 2\tilde{\alpha}_m\}\mathbf{E}\mathcal{D}_F(w^{(t)}, w^*)
\end{aligned}
$$

To simplify this inequality with condition $\tilde{\alpha}_m \leq \min\{\frac{\beta}{\sqrt{2}\dot{c}}, \frac{2}{\hat{\mu}_{F\cdot}}\}$, we have:

$$\boldsymbol{\Delta}^{(t+1)} \leq (1 - \frac{\tilde{\alpha}_m \hat{\mu}_{F\cdot}}{2})\boldsymbol{\Delta}^{(t)} + 6\tilde{\alpha}_m^2 \frac{N/S - 1}{N - 1}\sigma_{F,*}{}^2$$
$$+ \frac{2^{R+4}e\dot{c}}{\hat{\mu}_{F\cdot}\beta^2 R^2}[2(1 + 2R)\sigma_{F,*}^2 + \dot{\delta}]\tilde{\alpha}_m^3 + \frac{8\dot{\delta}}{\hat{\mu}_{F\cdot}}\tilde{\alpha}_m$$
$$- \{2 - \tilde{\alpha}_m[\frac{e(1 + \sigma_\Phi)\hat{L}_{\mathcal{E}}(\hat{u}_m + \eta\hat{\gamma}_\Phi)2^{R+6\frac{1}{2}}(\frac{1}{R} + 2)}{\hat{\mu}_{F\cdot}\beta R}$$
$$+ 12\frac{N/S - 1}{N - 1} + 3]\hat{L}_F\}\tilde{\alpha}_m\mathbf{E}\mathcal{D}_F(w^{(t)}, w^*)$$

where we use $3\tilde{\alpha}_m \leq \frac{6}{\hat{\mu}_{F\cdot}}$ and $\dot{c}\tilde{\alpha}_m \leq \sqrt{2}\beta(1 + \sigma_\Phi)\hat{L}_{\mathcal{E}}(\hat{u}_m + \eta\hat{\gamma}_\Phi)$

Let $\dot{c}_1 := 2 - \tilde{\alpha}_m[\frac{e(1+\sigma_\Phi)\hat{L}_{\mathcal{E}}(\hat{u}_m+\eta\hat{\gamma}_\Phi)2^{R+6\frac{1}{2}}(\frac{1}{R}+2)}{\hat{\mu}_{F\cdot}\beta R} + 12\frac{N/S-1}{N-1} + 6]\hat{L}_F$, and we have $\dot{c}_1 \geq 1$, when $\tilde{\alpha}_m$ satisfies:

$$\tilde{\alpha}_m \leq \hat{\alpha}_m := \frac{\hat{\mu}_{F\cdot}\beta R}{e(1 + \sigma_\Phi)\hat{L}_{\mathcal{E}}(\hat{u}_m + \eta\hat{\gamma}_\Phi)2^{R+6\frac{1}{2}}(\frac{1}{R} + 2) + 18(\hat{\mu}_{F\cdot}\beta R)\hat{L}_F}$$
$$\leq \frac{1}{[\frac{e(1+\sigma_\Phi)\hat{L}_{\mathcal{E}}(\hat{u}_m+\eta\hat{\gamma}_\Phi)2^{R+6\frac{1}{2}}(\frac{1}{R}+2)}{\hat{\mu}_{F\cdot}\beta R} + 12\frac{N/S-1}{N-1} + 6]\hat{L}_F} \tag{28}$$

By setting $\tilde{\alpha}_m$ with Equation ( 28), then let $\xi^{(t)} = (1 - \frac{\tilde{\alpha}\hat{\mu}_{F\cdot}}{2})^{-t-1}$ and $\mathcal{X}^{(T)} = \sum_{t=0}^{T-1}\xi^{(t)}$, $\tilde{\alpha}T \geq \frac{2}{\hat{\mu}_{F\cdot}}$, $\tilde{\alpha}_m \leq \min\{\frac{\beta}{\sqrt{2}\dot{c}}, \frac{2}{\hat{\mu}_{F\cdot}}\}$, we have:

$$\boldsymbol{\Delta}^{(t+1)} \leq (1 - \frac{\tilde{\alpha}_m\hat{\mu}_{F\cdot}}{2})\boldsymbol{\Delta}^{(t)} - \tilde{\alpha}_m\mathbf{E}\mathcal{D}_F(w^{(t)}, w^*) + \sum_{j=1}^{3}\dot{\delta}_j\tilde{\alpha}_m^j$$

where $\dot{\delta}_1 := \frac{8\dot{\delta}}{\hat{\mu}_{F\cdot}}$, $\dot{\delta}_2 := 6\frac{N/S-1}{N-1}\sigma_{F,*}{}^2$ and $\dot{\delta}_3 := \frac{2^{R+4}e\dot{c}}{\hat{\mu}_{F\cdot}\beta^2 R^2}[2(1 + 2R)\sigma_{F,*}^2 + \dot{\delta}]$.

Reformulate it as following:

$$\mathbf{E}\mathcal{D}_F(w^{(t)}, w^*) \leq \frac{1}{\tilde{\alpha}_m}[(1 - \frac{\tilde{\alpha}_m\hat{\mu}_{F\cdot}}{2})\boldsymbol{\Delta}^{(t)} - \boldsymbol{\Delta}^{(t+1)}] + \sum_{j=1}^{3}\dot{\delta}_j\tilde{\alpha}_m^{j-1}$$

Multiply both sides with $\xi^{(t)}$ and accumulate over $t$:

$$\mathbf{E}\mathcal{D}_F(\frac{\sum_{t=0}^{T-1}\xi^{(t)}w^{(t)}}{\mathcal{X}^{(T)}}, w^*) \leq \frac{\sum_{t=0}^{T-1}\xi^{(t)}}{\mathcal{X}^{(T)}}\mathbf{E}\mathcal{D}_F(w^{(t)}, w^*)$$
$$\leq \frac{1}{\tilde{\alpha}_m\mathcal{X}^{(T)}}\sum_{t=0}^{T-1}[(1 - \frac{\tilde{\alpha}_m\hat{\mu}_{F\cdot}}{2})\xi^{(t)}\boldsymbol{\Delta}^{(t)} - \xi^{(t)}\boldsymbol{\Delta}^{(t+1)}] + \sum_{j=1}^{3}\dot{\delta}_j\tilde{\alpha}_m^{j-1}$$
$$= \frac{1}{\tilde{\alpha}_m\mathcal{X}^{(T)}}\boldsymbol{\Delta}^{(0)} - \frac{\xi^{(T-1)}}{\tilde{\alpha}_m\mathcal{X}^{(T)}}\boldsymbol{\Delta}^{(T)} + \sum_{j=1}^{3}\dot{\delta}_j\tilde{\alpha}_m^{j-1}$$
$$= \frac{\hat{\mu}_{F\cdot}}{2\xi^{(T-1)}[1 - (1 - \tilde{\alpha}_m\hat{\mu}_{F\cdot}/2)^T]}\boldsymbol{\Delta}^{(0)} - \frac{\xi^{(T-1)}}{\tilde{\alpha}_m\mathcal{X}^{(T)}}\boldsymbol{\Delta}^{(T)} + \sum_{j=1}^{3}\dot{\delta}_j\tilde{\alpha}_m^{j-1}$$
$$\leq \hat{\mu}_{F\cdot}e^{-\tilde{\alpha}_m\hat{\mu}_{F\cdot}T/2}\boldsymbol{\Delta}^{(0)} - \frac{\hat{\mu}_{F\cdot}}{2}\boldsymbol{\Delta}^{(T)} + \sum_{j=1}^{3}\dot{\delta}_j\tilde{\alpha}_m^{j-1}$$
$$\leq \hat{\mu}_{F\cdot}e^{-\tilde{\alpha}_m\hat{\mu}_{F\cdot}T/2}\boldsymbol{\Delta}^{(0)} + \sum_{j=1}^{3}\dot{\delta}_j\tilde{\alpha}_m^{j-1}$$
$$\leq \mathcal{O}[\mathcal{D}_F(\bar{w}^{(T)}, w^*)]$$

where $\bar{w}^{(T)} := \frac{\sum_{t=0}^{T-1} \xi^{(t)}}{\mathcal{X}^{(T)}} w^{(t)}$, we use convexity of $\mathcal{D}_F$ and $F$ for the first inequality, the second one is by the reformulated inequality and the third one is by setting $\tilde{\alpha}_m T \geq \frac{2}{\hat{\mu}_{F.}}$ and the fact $\frac{2\xi^{(T-1)}}{\tilde{\alpha}_m \hat{\mu}_{F.}} \geq$ $\mathcal{X}^{(T)} = \frac{2\xi^{(T-1)}[1-(1-\frac{\tilde{\alpha}_m \hat{\mu}_{F.}}{2})^T]}{\tilde{\alpha}_m \hat{\mu}_{F.}} \geq \frac{\xi^{(T-1)}}{\tilde{\alpha}_m \hat{\mu}_{F.}}$ and $0 \leq (1 - \frac{\tilde{\alpha}_m \hat{\mu}_{F.}}{2})^T \leq e^{-\frac{1}{2}\tilde{\alpha}_m \hat{\mu}_{F.} T} \leq e^{-1} \leq \frac{1}{2}$.

To tighten this bound, we recommend [4], which discusses the range and strategy of step sizes in detail rather than our unified bound.

With $\tilde{\alpha}_m \geq \frac{2}{\hat{\mu}_{F.} T}$, we have:

$$\sum_{j=1}^{3} \dot{\delta}_j \tilde{\alpha}_m^{j-1} \leq \mathcal{O}(\dot{\delta}_1) + \mathcal{O}(\frac{\dot{\delta}_2}{T\hat{\mu}_{F.}}) + \mathcal{O}(\frac{\dot{\delta}_3}{T^2 \hat{\mu}_{F.}^2})$$

Thus,

$$\mathcal{O}[\mathcal{D}_F(\bar{w}^{(T)}, w^*)] = \mathcal{O}(\hat{\mu}_{F.} e^{-\tilde{\alpha}_m \hat{\mu}_{F.} T/2} \mathbf{\Delta}^{(0)}) + \mathcal{O}(\frac{\dot{\delta}}{\hat{\mu}_{F.}})$$
$$+ \mathcal{O}(\frac{(N/S-1)\sigma_{F,*}^2}{NT\hat{\mu}_{F.}}) + \mathcal{O}(\frac{2^{R+4} e\dot{c}}{T^2 \hat{\mu}_{F.}^3 \beta^2 R^2}[2(1+2R)\sigma_{F,*}^2 + \dot{\delta}])$$

where, $\dot{\delta} = 4[\lambda\frac{\hat{L}_{g*}}{\hat{\mu}_{F.}}(\hat{u}_m + \eta\hat{\gamma}_\Phi)]^2(\frac{\hat{\gamma}_f^2}{|\tilde{d}_i|} + \hat{\epsilon}^2) + 16[(1+\sigma_\Phi)\hat{L}_\mathcal{E}(\hat{u}_m + \eta\hat{\gamma}_\Phi)\hat{\gamma}_\Phi]^2$ and $\dot{c} = 4[(1+\sigma_\Phi)\hat{L}_\mathcal{E}(\hat{u}_m + \eta\hat{\gamma}_\Phi)]^2$.

For simplification, letting $A = [\frac{\hat{L}_{g*}}{\hat{\mu}_{F.}}(\hat{u}_m + \eta\hat{\gamma}_\Phi)]^2(\frac{\hat{\gamma}_f^2}{|\tilde{d}_i|} + \hat{\epsilon}^2)$, $B = [(1+\sigma_\Phi)\hat{L}_\mathcal{E}(\hat{u}_m + \eta\hat{\gamma}_\Phi)\hat{\gamma}_\Phi]^2$ and $C = \frac{\sigma_\Phi^2 \hat{L}_\mathcal{E}^2(\hat{u}_m + \eta\hat{\gamma}_\Phi)^2}{\hat{\mu}_{F.}^3}$, we have:

$$\mathcal{O}[\mathcal{D}_F(\bar{w}^{(T)}, w^*)] = \mathcal{O}(\hat{\mu}_{F.} e^{-\tilde{\alpha}_m \hat{\mu}_{F.} T/2} \mathbf{\Delta}^{(0)}) + \mathcal{O}(\frac{A\lambda^2 + B}{\hat{\mu}_{F.}})$$
$$+ \mathcal{O}(\frac{(N/S-1)\sigma_{F,*}^2}{NT\hat{\mu}_{F.}}) + \mathcal{O}(\frac{2^R C}{T^2 \beta^2 R^2}[R\sigma_{F,*}^2 + A\lambda^2 + B]).$$

$\square$

### D.7.2 Proof of Theorem 2

The proof of Theorem 2 is shown as followings:

*Proof.* With Gaussian prior and first-order methods, we have the bound between personalized model and optimal global model, with $\dot{\delta}_p = \frac{2}{\hat{\mu}_{F_{i,\cdot}}^2}(\frac{\hat{\gamma}_f^2}{|\tilde{d}_i|} + \hat{\epsilon}^2) + \frac{2}{\lambda^2}\epsilon_1^2 + \frac{4}{\lambda^2}\sigma_{F,*}^2 + \frac{1}{2}\eta^2\mathcal{G}_\Phi^2$, and $\dot{c}_p = (\frac{32}{\lambda^2}\hat{L}_F + \frac{8}{\hat{\mu}_{F.}})$:

$$\mathbf{E}||\tilde{\theta}_i(\bar{w}^T) - w^*||^2 \leq 4[\mathbf{E}||\tilde{\theta}_i(\bar{w}^T) - \theta_i^*(\bar{w}^T)||^2$$
$$+ \mathbf{E}||\theta_i^*(\bar{w}^T) - \mu_i(\bar{w}^T)||^2 + \mathbf{E}||\mu_i(\bar{w}^T) - w^{(T)}||^2 + \mathbf{E}||w^{(T)} - w^*||^2]$$

$$\leq 4[\frac{2}{\hat{\mu}_{F_{i,\cdot}}^2}(\frac{\hat{\gamma}_f^2}{|\tilde{d}_i|} + \hat{\epsilon}^2) + \frac{1}{\lambda^2}\mathbf{E}2[||\nabla\tilde{F}_i(w^{(T)}) - \nabla F_i(w^{(T)})||^2$$
$$+ ||\nabla F_i(w^{(T)})||^2] + \frac{1}{2}\mathbf{E}||\eta\nabla\Phi_i(w^{(T)})||^2 + \mathbf{E}||w^{(T)} - w^*||^2]$$

$$\leq 4[\frac{2}{\hat{\mu}_{F_{i,\cdot}}^2}(\frac{\hat{\gamma}_f^2}{|\tilde{d}_i|} + \hat{\epsilon}^2) + \frac{2}{\lambda^2}\mathbf{E}\{\epsilon_1^2 + 2[||\nabla F_i(w^{(T)}) - \nabla F_i(w^*)||^2 + ||\nabla F_i(w^*)||^2]\}$$
$$+ \frac{1}{2}\eta^2\mathcal{G}_\Phi^2 + \mathbf{E}||w^{(T)} - w^*||^2]$$

$$\leq 4[\frac{2}{\hat{\mu}_{F_{i,\cdot}}^2}(\frac{\hat{\gamma}_f^2}{|\tilde{d}_i|} + \hat{\epsilon}^2) + \frac{2}{\lambda^2}\epsilon_1^2 + \frac{8}{\lambda^2}\hat{L}_F\mathcal{D}_F(w^{(T)}, w^*) + \frac{4}{\lambda^2}\sigma_{F,*}^2$$
$$+ \frac{1}{2}\eta^2\mathcal{G}_\Phi^2 + \mathbf{E}||w^{(T)} - w^*||^2]$$

$$\leq 4[\frac{2}{\hat{\mu}_{F_{i,\cdot}}^2}(\frac{\hat{\gamma}_f^2}{|\tilde{d}_i|} + \hat{\epsilon}^2) + \frac{2}{\lambda^2}\epsilon_1^2 + \frac{4}{\lambda^2}\sigma_{F,*}^2 + \frac{1}{2}\eta^2\mathcal{G}_\Phi^2 + (\frac{8}{\lambda^2}\hat{L}_F + \frac{2}{\hat{\mu}_{F_\cdot}})\mathcal{D}_F(w^{(T)}, w^*)]$$

$$\leq \mathcal{O}(\dot{\delta}_p) + \mathcal{O}[\dot{c}_p\mathcal{D}_F(\bar{w}^{(T)}, w^*)]$$

where the first inequality is by Proposition 3, the second one is by Lemma 1 and Proposition 3, the third one is by Assumption 8 and Lemma 5, the fourth one is by Lemma 2 and Assumption 7 and the final one is by Theorem 1. □