# OpenReview forum: "PRIOR: Personalized Prior for Reactivating the Information Overlooked in Federated Learning."
_NeurIPS.cc/2023/Conference — NeurIPS 2023 poster_

### Official Review · Reviewer_JezB · 2023-06-13

**Soundness:** 3 good
**Presentation:** 3 good
**Contribution:** 3 good
**Rating:** 6
**Confidence:** 3

**Summary:**


This paper targets injecting personalized prior knowledge into the global model, which attempts to mitigate the introduced incomplete information problem in PFL. The idea is to decouple the personalized prior from the local objective function regularized by Bregman divergence. The mirror descent (RMD) is used to extract the prior.
The authors presented convergence analysis and showed many experimental results.


**Strengths:**

1. The authors conducted many experiments and show improved results. For example, the proposed method has a higher deviation in missing classes in local testing and a lower deviation in global testing.
2. The idea seems to be interesting.

**Weaknesses:**

1. Unclear motivation and literature review.
2. Lack of an explainable summary of the main idea. Given the abstract, I think the idea seems to be interesting, but I got lost in the details and symbols when reading Section 4 and Section 5.
3. The writing requires to be improved.


**Questions:**

**Background**
1. In Lines 43-46, "Most of the insightful works [17, 50] propose assumptions for recovering this incomplete information, but these assumptions are implicit, which limits the way to use the information to develop personalized strategies. To address the former issue above, ...".
What is the exact problem/issue of the previous assumption? Can the authors explain more about the motivation? What is "limits the way to use the information" exactly here? I did not catch the exact challenge that the authors target here.

2. In lines 46,47, the major contribution comes from "we propose framework pFedBreD to inject personalized prior knowledge (PPK) into the one provided by a global model." Is there a research line for injecting personalized prior knowledge? What is the advantage of pFedBreD compared with them?
I see the authors elaborated on "Ablation Analysis of Personalized Prior" from Line 269 to Line 283. Can authors explain more about PPK-relevant research line in a more intuitive manner?

**Method and Framework**

3. In lines 103,104, "Exponential Family The regular exponential family (X-family) is a relatively large family... Therefore, to yield the prior, we employ the X-family..." Is the employment of X-family due to "large family"? Is there any other special advantage for X-family?

4. I can follow the equations of Section 4 from a math view. Would the authors elaborate the intuitive logic of combining these equations?

5. Section 4 is "methodology", meanwhile, Section 5 is "framework". Does the designed framework belong to the proposed methodology? Or, what is the relation between the two sections?

6. In Line 156, "Inspired by the aforementioned motivation," What is the "motivation"? I tried to search "motivation," but "motivation" appears only once in the paper...What is the "motivation" here? Is it relevant to some types of math objective functions?

7. In Line 163 and Line 164, "To solve the optimization problem in Eq. (12), we use gradient-based methods to solve the global problem ..." Based on my knowledge, the conventional method is "gradient-based methods." Is there any other special optimization in your framework?

**Experiment**

8. In Line 220, "The results of average accuracy per client are shown in Table 1." Could the authors summarize the insight conclusion of Table 1 in the main text?

9. What is the relation between "RMD" and "mh"? Are any experiments relevant to "RMD"?

**Minor**

- In Line 85, "a expectation"
- The clickable indices of reference/table/equation/etc do not work, which are not convenient for searching the relevant contents.
- In Line 113 - Line 115, long sentence with grammar error.

  "In this section2, we introduce missing client-sampling information based on classic FL use EM to reduce the computational cost of the information-introduced FL problem, and propose RMD, a class of prior selection strategies, based on the E-step in EM"
- Equation 5 requires to introduce "KL" function before usage.
- In Line 168, T,R,N appear suddenly by following the main body.


**Limitations:**

The authors try to empirically address the limitation of instability in the global model with aggregation noise.

---

> ### Author Rebuttal · Authors · 2023-08-09
>
> ***Comments***:
> - We sincerely thank reviewer JezB for appreciating our idea. The main concern of the reviewer is about main idea and our methodology, and we answer the questions one by one and make the structure and design of our paper clear.
>
> ***Responses***:
> 1. [W1, W2, and W3] Motivation and readability:
> - We set out the *main idea and structure* of the paper in short. This paper aims to address the *information overlooked problem* and all the followings are proposed for this. Please see [general responses](https://openreview.net/forum?id=kuxu4lCRr5&noteId=WzbbMONDyW) for detailed explanation about motivation [W1, W2] and readability improvement [W3].
> 2. [Q1, W1] The issue of the implicit assumption? How does it limit the way?
> - The main purpose of the explicit expressions in most classical methods is for parameterization, computational convenience, and closed-form solutions, e.g., re-parameterization for variational inference, Bayesian conjugate prior.[f, g]
> - In this paper, under implicit prior assumption, the complete information is not parameterized and computationally expensive in Line 125-134. The substitute is to tune this information by the way of regularization, representation or loss function design. Under explicit prior assumption, if the parameters are given, the prior distribution is *completely determined*, and the local information that comes after that, *comes exclusively from the local data*. Thus, we can design theoretically supported strategies and directly with parameters.
> 3. [Q2, W1] Is there a research line for injecting personalized prior knowledge (PPK)?
> - To the best of our knowledge, this is *the first paper* presenting this concept of PPK in PFL. The introduction of PPK is to address the problem of overlooked information, and this is the first time this problem has been formally discussed in PFL. For space reason, we can discuss this in the discussion period.
> 4. [Q3] Why X-family? More discussion.
> - In order to expand the discussion of the main problem in this paper, we use the *X-family prior assumption*, for its *broad coverage* in both practice and theoretical analysis. According to the *relationship between X-family and B-Div*, as shown in Equation 3 and more in Line 601-612 in Appendix, the introduction of the B-Div brings about the properties of *easy calculation*, e.g., the first-order moment estimation point is the expected point with the closest Bregman distance (i.e. B-Div), in Line 507-509 in Appendix. the properties bring many *closed-form solutions that can be computed by replacing numerical approximations*, e.g., the gradient of Bregman-Moreau envelope, the simpler form of both the Fenchel conjugate duality and transformation of the natural and expected parameters of an X-family. This assumption makes the problem well-computable in Line 127-139 with optimization framework, and the computation method mirror descent is introduced with theoretical support [39]. If there is anything unclear in our responses, we'd like to discuss in the discussion period.
> 5. [Q4] The intuitive logic of the equations in Section 4, as follows [Please see the motivation explanation in [general responses](https://openreview.net/forum?id=kuxu4lCRr5&noteId=WzbbMONDyW)]:
>   - Math. modeling. [Eq. 5]
>   - Implicit complete information $\Theta_{i}$ is *not directly computable*. [Eq. 6]
>   - Expectation maximization (EM) *makes it computable* and the prior explicit. [Eq. 7-8]
>   - The optimization method Mirror Descent (MD) is used *to compute the EM problem*. [Eq. 9]
>   - Relaxing the constrains of MD so that there is *room for personalization*. [Eq. 10-11]
> 6. [Q5] The relation between Sec. 4 and 5.
> - Section 4 is about *optimization* methodology, and Section 5 is about *computation* framework. The latter is a practical implementation of the former.
> 7. [Q6] What does the *aforementioned motivation* in Line 156 stand for?
> - Typical motivation for a computation framework is *the modeled optimization problem*, which is in Line 148-153, the methodology in Section 4.
> 8. [Q7] Why is *gradient-based methods* mentioned?
> - We did *not fix* exactly how to solve the bi-level optimization problem *until Section 5*. There are many computation methods for the optimization problem. In Section 5, we only employ gradient-based method, and other methods are also considerable, e.g., Newton, evolutionary, MCMC-Bayesian or other Approximate-Bayesian sampling-based computation methods discussed in Line 540-570 in Appendix.
> 9. [Q8] The insightful conclusion of Table 1.
> - Table 1 is for comparison *in Line 241-262*, we provide insights from 3 perspectives, including convex or non-convex problems, easy or hard tasks, and text tasks. The most insightful one we think is *in Line 257-262*: the overlooked information makes the specific model on each client have to re-obtain this information from scratch solely from the data during training. Table 1 provides an overall evaluation and the tasks our method specializes in as claimed *in Line 65-67*.
> 10. [Q9] What is the relation between "RMD" and "mh"? Are any experiments relevant to "RMD"?
> - RMD is a class of strategies determined by $\Phi$.
> - With given $\Phi = f_{i}+F_{i}$, mh is an implementation of RMD, or a strateiges in the class of strategies RMD.
> - Any experiments about mh are the experiments relevant to RMD.
>
> ***Minors***:
> We will carefully check the paper to correct similar issues.
> 1. Yes, that should be *an*.
> 2. Technical issue. We fix it.
> 3. The sentence should be *...on classic FL, use EM...*. [missed comma]
> 4. KL-divergence will be elaborated.
> 5. We will make it better for reading.
>
> ***References***:
>
> [f] Variational Autoencoder Based Anomaly Detection Using Reconstruction Probability. Special lecture on IE 2.1 (2015): 1-18.
>
> [g] A geometric View of Conjugate Priors. ML 81 (2010): 99-113.

---

> > ### Comment · Reviewer_JezB · 2023-08-11
> >
> > Thanks for the detailed response and patient explanation. I will improve my rating but I need more time to consider everything on this page. I will give an additional comment later.

---

> > > ### Author Response · Authors · 2023-08-12
> > > **Response to the Official Comment**
> > >
> > > Thanks for participating actively in the discussion and we sincerely appreciate the time and effort the reviewer JezB put into this work. We are delighted that our responses address the concerns and are perceived positively.
> > >
> > > Please feel free to engage in discussion with us during the feedback phase, as we are eager to receive constructive suggestions to enhance our work.

---

> > > > ### Comment · Reviewer_JezB · 2023-08-13
> > > >
> > > > Hi, thanks for the authors' efforts. My concerns have been solved. I have re-read the paper + rebuttals before updating my score.
> > > >
> > > > My rating comes from the paper + the rebuttal + paper ranking (after rebuttal) in my batch.

---

> > > > > ### Author Response · Authors · 2023-08-13
> > > > > **Response to the Official Comment**
> > > > >
> > > > > Thanks for the positive feedback and appreciation. We are glad that the concerns have been solved.

---

### Official Review · Reviewer_qY3t · 2023-07-05

**Soundness:** 3 good
**Presentation:** 2 fair
**Contribution:** 3 good
**Rating:** 7
**Confidence:** 3

**Summary:**

The authors of the paper consider the problem of personalized federated learning. The main problem the authors attempt to tackle is the information on sampling of clients being overlooked. Specifically, they attempt to introduce two major steps in the training of personalized models: the first one is the injection of personalized prior knowledge into the global model before training, the second is the extraction of the prior before sending the global model updates. The authors formally present the problem of overlooking client-sampling information, then they utilize a framework for the problem called pFedBreD. Additionally, they provide extensive theoretical examination of the framework, as well as numerous experiments to validate the superior performance of their solution compared to a handful of baseline algorithms.

**Strengths:**

- The paper is very well-written. It is clear and comprehensive, and its structure is sound.
- The authors motivate the problem very well, and provide a sound formulation of it.
- The theoretical examination of the problem, especially the results are well-presented.
- The extensive experimental results, especially the comparisons with various baselines are useful for the completeness of the evaluation of the proposed framework. A number of datasets as well as different models are considered. Moreover, the analysis of the results is insightful.

**Weaknesses:**

I have a couple of comments on the weaknesses of the paper:
- I think the main body of the paper is well-written as I mentioned in the weaknesses, but I also think it lacks fundamental parts that can be added to improve the readability of the paper. For example, the pFedBreD algorithm is not included in the main body but it is an important part that needs to be presented in the main body of the paper for completeness of the presentation. Similarly, the authors refer to Theorem 1 in the paper but it is not mentioned until the Appendix D. The paper should be self sufficient for the reader.
- The authors use acronyms in the paper before mentioning what they are, for example the use of **mh**, **lg**, and **meg** before explicitly defining them or mentioning what they stand for. This can be confusing for the reader and needs to be rectified.

**Questions:**

The mentioned weaknesses need to be addressed. The main body of the paper should be improved to include the missing parts as well as the acronyms should be clearly defined.

Further, the paper needs to be checked for typos. For example: line 85, "pFedBreD is a expectation" -> "pFedBreD is an expectation", line 95 "Bregman divergencee" -> "Bregman divergence", line 213: "we choose following" -> "we choose the following".

**Limitations:**

The authors present the limitations of their work.

---

> ### Author Rebuttal · Authors · 2023-08-09
>
> ***Comment***:
> - We sincerely thank the reviewer qY3t for the appreciation and the constructive suggestion to further improve clarity and readability.
>
> ***Responses***:
> 1. [W1, Q1] Suggestions for improving readability.
> - Thanks for the constructive suggestion for readability, we will do the following modifications in the revision:
>   - The pFedBreD algorithm in Line 527-530 in Appendix A.5 will be put between Line 169 and Line 170 in the main body of the paper from the appendix.
>   - The Theorems in Appendix D.1 will be put in Section 5 before Remark 1 in the main body.
> - Other modifications for readability are listed as follows: (Thanks for all the suggestions from the reviewers)
>   - All the full names of the acronyms will be placed before or around the acronyms. [W2, which will be discussed in details later]
>   - The motivation and framework illustration, Fig. 1, will be more visually linked to the various relevant parts of the math. methodology and computation framework. [Suggested by reviewer [Q28a](https://openreview.net/forum?id=kuxu4lCRr5&noteId=uuT3KKlAUX)]
>   - We will add a subsection to formalize the overlooked information problem, focusing on the followings: [Suggested by reviewer [jGgs](https://openreview.net/forum?id=kuxu4lCRr5&noteId=XWZD1FWgxw)]
>        - **The core of the problem**: global model $w=E_{i}w_{i}=E_{i}w_{i}|i \Rightarrow$ Mutual Information $I(w;i)=0$.
>        - **Effects**: global model *has no mutual information with client-sampling $i$*, as shown in the above equations and discussed in Line 120-124, in particular when applying regularization $R(w^{(t)};...)$ or local initialization $w_{i,0}^{(t)}\leftarrow w^{(t)}$  where $w_{i}$ is the local model on the $i^{th}$ client. This makes the specific model on each client have to re-obtain this information from scratch solely from the data during training which is particularly distinctive on text tasks as the analysis in Line 259-262, and especially  impacted on hard-to-learn representations and datasets.
>
> 2. [W2, Q1] The unclearly defined parts of acronyms.
> - Thanks for pointing out. All the full names of the acronyms will be placed before or around the acronyms. *In Line 634-638 in the Appendix*, we give the definitions and reasons about the full names of the acronyms. The three implementations of $\mu_{i}$, i.e. lg, meg and mh, represent *loss gradient*, *memorized envelope gradient* and *memorized hybrid* respectively. *Memorized* means that we choose the gradient of Bregman-Moreau envelope  $\nabla F_{i}(w_{i,r-1}^{(t)})$ as $\eta[w_{i,R}^{(t-1)} - \theta_{i,r-1}^{(t)}]$, where $\eta \ge 0$ is a step-size-like hyper-parameter. Each local client memorizes their own local part of the latest global model $w^{(t)}$ at the last global epochs $w_{i,R}^{(t-1)}$, instead of $w_{i,r-1}^{(t)}$ in practice.
>
> ***Minors***:
> 1. [Q2] Typos.
> - Thanks for pointing out, and we will fix all typos in the revision.

---

> > ### Comment · Reviewer_qY3t · 2023-08-12
> >
> > I have read the other reviews and the detailed responses by the authors. I thank the authors for addressing my comments, and other reviewers', accordingly, and highlighting such revisions in the response.

---

> > > ### Author Response · Authors · 2023-08-12
> > > **Response to the Official Comment**
> > >
> > > We would like to express our heartfelt appreciation to the reviewer qY3t for the constructive comments and acknowledgment in reviewing and improving our work.

---

### Official Review · Reviewer_jGgs · 2023-07-06

**Soundness:** 3 good
**Presentation:** 2 fair
**Contribution:** 3 good
**Rating:** 6
**Confidence:** 4

**Summary:**

This paper proposes pFedBreD, which decouples the personalized prior from the local objective function regularized by Bregman divergence for greater adaptability in personalized FL. Extensive experiments validate the effectiveness of the proposed method on 5 datasets.

**Strengths:**

S1. The problem of overlooking client-sampling information at prior knowledge being transferred is important.
S2. A novel framework, pFedBreD, is proposed for computing the Bayesian optimization problem. The theorem is provided to analyze the convergence.
S3. The experiments validate the effectiveness of pFedBreD on five widely-used datasets.


**Weaknesses:**

W1. The problem is not well-motivated. To be more specific, the shortcomings of existing PFL methods are not well explained. The client-sampling information should be clearly illustrated, and the reason why the prior knowledge extraction is challenging is missing.

W2. The problem should be formally defined. It would be better to add a subsection to provide the statement of the research problem of this paper.

W3. The efficiency of the proposed methods is not well analyzed.


**Questions:**

Q1. The assumption this paper makes is the uniform sampling of clients. How about other sampling methods? Is the proposed method applicable to other sampling scenarios?

Q2. In Table 1, the accuracy of FedAMP is better than the proposed method on some datasets. In particular, the accuracy of the MCLR of FedAMP on different datasets is close to or even outperforms mh. The reason should be provided.

Q3. How to determine the number of global/local epochs in your experiment setting?


**Limitations:**

The authors adequately pointed out the limitations of this work.

---

> ### Author Rebuttal · Authors · 2023-08-09
>
> ***Comment***:
> - We sincerely thank the reviewer jGgs for appreciation recognizing the importance of the problem we introduced and suggestion for improving readability. We address the concern raised by the reviewer about motivations and challenges, and we answer these questions as follows.
>
> ***Responses***:
>
> 1. [W1] More explanation about motivations.
> - Thank you for pointing out. We provide further explanation about the existing shortcomings in PFL that this paper is focusing on, and the two major shortcomings are as follows [Please see [general responses](https://openreview.net/forum?id=kuxu4lCRr5&noteId=WzbbMONDyW) for more about motivations]:
>    1. **What is exactly the effects of overlooked client-sampling information?**: the salient shortcoming of this overlooking is that *the specific global knowledege for each client is the same*, making the specific model on each client have to re-obtain the local information from scratch solely from the data during training which is particularly distinctive in text tasks as analyzed in Line 259-262. These are especially important on hard-to-learn representations and datasets.
>    2. **Why is the prior knowledge *explicit* extraction challenging?**: the main purpose of the explicit expressions in most classical methods is for parameterization, computational convenience, and closed-form solutions, e.g., variational inference for re-parameterization techniques, conjugate prior in Bayesian methods. In this paper, under implicit prior assumption, the complete information is *not parameterized and computationally expensive* in Line 125-134. The substitute is to tune this information by the way of regularization, representation or loss function design. Under explicit prior assumption, if all the parameters are given, the prior distribution is *completely determined*, and the local information that comes after that, *comes exclusively from the local data*. Thus, we can design theoretical-supported strategies directly with parameters.
>
> 2. [W2] Formal definition and suggestion about additional subsection.
> - Thanks for the suggestion about a subsection about motivation to improve reading with formal definition of the overlooked information problem. We consider adding sections focusing on the followings: [Welcome to discuss in detail during the discussion period.]
>    - **The core of the problem**: global model $w=E_{i}w_{i}=E_{i}w_{i}|i \Rightarrow$ Mutual Information $I(w;i)=0$.
>    - **Effects**: global model *has no mutual information with client-sampling $i$*, as shown in the above equations and discussed in Line 120-124, in particular when applying regularization $R(w^{(t)};...)$ or local initialization $w_{i,0}^{(t)}\leftarrow w^{(t)}$  where $w_{i}$ is the local model on the $i^{th}$ client. This makes the specific model on each client have to re-obtain this information from scratch solely from the data during training which is particularly distinctive in text tasks as the analysis in Line 259-262.
>
> 3. [W3] More analysis about efficiency.
> - Thanks for suggestion, and we will discuss more about efficiency as follows and in revision. Please see [Q28a rebuttal Q3](https://openreview.net/forum?id=kuxu4lCRr5&noteId=KKUTOVP2PR) for more details, and we'd like to provide more during the discussion period:
>   - We discuss *efficiency* in terms of time and space complexity. The complexity comparison is shown below, which is *in Appendix A and the last column in Table 2*. Our method *share the same complexity* as pFedMe and FedAMP, where $N$,$T$,$R$ and $K$ are respectively the number of clients, global epochs, local epochs and proximal solver iterations. *As shown below and in Table 2*, the local computation only adds limited computation with significant improvement.
>
>  |Complexity/Methods|FedAvg|pFedMe|Per-FedAvg|FedAMP|pFedBreD (ours)|
>  |-|-|-|-|-|-|
>  |Entire Sys. Memory|$O(N)$|$O(N)$|$O(N)$|$O(N)$|$O(N)$|
>  |Entire Sys. Time|$O(NTR)$|$O(NTRK)$|$O(NTR)$|$O(NTRK)$|$O(NTRK)$|
>  |Additional Local Computation beyond FedAvg|-|Addition|Gradient|Weighted Sum and L2-Distance|Gradient and Addition|
>  |Average Acc. on All Tasks|72.80|79.48|80.02|81.80|83.09|
>
> 4. [Q1] How about other sampling scenarios?
> - The proposed method *is applicable to any scenarios* with a known or given sampling distribution if the expectation exists, as we mentioned *in Line 188-189* in Convergence Analysis (CA). We analyze on a *uniform* client sampling $E_{i}=\frac{1}{N}\sum_{i=1}^{N}$ setting for simplification. *Other sampling methods can be obtained* with client sampling expectation $E_{i} [F_{i}] = F$, by changing sampling weights. In the CA, we use the uniform sampling *only for simplification* of the theorem, and only the linearity of operator $\frac{1}{N}\sum_{i=1}^{N}$ in convexity, Jensen inequality and gradient calculation, which could be replaced by linear $E_{i}$.
>
> 5. [Q2] Analysis about FedAMP.
> - We do provide an analysis about FedAMP *in Line 242-251*, which is fully consistent with our experimental results, and the main analysis provided are as follows:
> 	-  *On convex problem, FedAMP outperforms our method ... One possible reason ... FedAMP uses the distance between models as a similarity in the penalty point selection ... since there is only one global optimum, this penalty point tends not to change ... a non-dynamic regular term ... will not be as advantageous for non-convex problems ... as penalty point tends to fall into the local optimum ...*
>
> 6. [Q3] The number of global/local epochs.
> - The number of global/local epochs $T$/$R$ are shown below and in Line 224. Empirically, different tasks require different numbers of training epochs to reach convergence, and it's recommended that smaller $R$ if global stability is wanted, and larger if personalization and convergence speed are wanted.
>
> | Dataset-Model| CIFAR-10 | Sent140-LSTM | FEMNIST-MCLR/DNN | FMNIST-MCLR/DNN | MNIST-MCLR/-DNN |
> |-|-|-|-|-|-|
> |Local Epochs $R$|20|20|20|20|20|
> |Global Epochs $T$|2000|800|800|200|200|

---

> > ### Comment · Reviewer_jGgs · 2023-08-12
> >
> > Thanks for the response. I believe that the authors have addressed all my concerns and I will keep the score.

---

> > > ### Author Response · Authors · 2023-08-12
> > > **Response to the Official Comment**
> > >
> > > We would like to express our sincere gratitude to the reviewer for endorsing our work and providing constructive suggestion.

---

### Official Review · Reviewer_HD18 · 2023-07-07

**Soundness:** 2 fair
**Presentation:** 3 good
**Contribution:** 2 fair
**Rating:** 5
**Confidence:** 3

**Summary:**

In this paper, the authors propose pFedBreD to decouple prior knowledge from each client. pFedBreD extracts the personalized prior with Bregman Divergence for better performing personalized tasks. The authors provide convergence analysis and experiments evaluated on 5 datasets.

**Strengths:**

1.    The authors give a detailed convergence analysis for the proposed method.
2.    The theoretical derivation of the paper is comprehensive in the Appendix.
3.    The authors provided extensive experiments for different tasks and models and the results show the effectiveness of improving the local models' performance.


**Weaknesses:**

1.    The motivation of the work is not clearly presented. It is hard to understand why the authors use Bregman Divergence and Relaxing Mirror Descent.
2.    It is confusing that the authors use the average results and the standard deviation of them on all tasks to represent the overall performance of methods. Could the authors give some intuition or explain the purpose of this?
3.    More related works need to be introduced. And we'd like to see experimental results compared to more related works, such as [1], [2], [3], and [4].
4.    This paper focuses on the specific label distribution problem. It would be better if the authors could discuss the performance of other heterogeneous data distribution problems (e.g., different number of classes is different).

[1] Personalized Federated Learning with Feature Alignment and Classifier Collaboration. ICLR 2023.
[2] Ditto: Fair and robust federated learning through personalization. ICML 2021.
[3] Personalized Federated Learning with First Order Model Optimization. ICLR 2021.
[4] Personalized federated learning using hypernetworks. ICML 2021.

**Questions:**

1.    The intuition of the work is not clear. Could the authors explain the motivation for using Bregman Divergence and Relaxing Mirror Descent in detail?
2.    In Table 1, why the second-best performance is 79.44 (mh) rather than 79.68 (Per-FedAvg+FT)?

**Limitations:**

The authors have discussed the potential limitations.

---

> ### Author Rebuttal · Authors · 2023-08-09
>
> ***Comment***:
> - We sincerely thank the reviewer HD18 for the appreciation and constructive comments. The reviewer raises concern about the motivation of using Bregman-Divergence (B-Div) and relaxing Mirror Descent (RMD), and we answer the questions one by one.
>
> ***Responses***:
> 1. [Q1, W1] The Motivation of B-Div and RMD.
> - **B-Div**:
>   - **Modeling**: the main problem studied in this paper is the overlooked information of client-sampling. In order *to widen the range of the main problem math. modeling*, we use the exponential family (X-family) prior assumption, due to its broad coverage in both practice and theoretical analysis.
>   - **Calculation**: according to the relationship between X-family and B-Div, as shown in Equation 3 and more in Line 601-612 in Appendix, the introduction of the B-Div brings about *the properties of easy calculation*, e.g., the first-order moment estimation point is the expected point with the closest Bregman distance (i.e., B-Div), in Line 507-509 in Appendix.
>   - **Computation**: these properties bring many easy-compute *closed-form solutions* instead of numerical approximations, e.g., the gradient of Bregman-Moreau envelope, the simpler form of both the Fenchel conjugate duality and transformation of the natural and expected param. of an X-family.
> - **RMD**:
>   - **Pre**：the introduction of Expectation Maximization (EM) and the X-family prior assumption in Line 124-134 transforms the optimization problem into an easy-compute one, from implicit into explicit assumption.
>   - **Mirror descent (MD)**: we use MD as the proposed computational framework with rigor, according to the relationship among EM, X-family and MD, i.e., EM under the X-family assumption is MD [39] in Line 142.
>   - **Relaxing**: we relax the constrains of MD *provides space for personalization* that MD would not otherwise have, in Line 140-148.
> 2. [W2] Why do we put the avg. and the std. on all tasks in Tables?
> - Thanks for pointing out. Since we have a lot of benchmarks and *for ease-to-read*, to catch the overall results of the comparison, we provide the additional columns to provide statistics of the performance and stability. It's better to *differentiate between hard tasks and others*.
> 3. [W3] *More related works and comparison*.
> - Thanks for the constructive suggestions. We add the comparison with the methods [a-d] in the revision. We provide the results in [*GENERAL RESPONSES*](forum?id=kuxu4lCRr5&noteId=WzbbMONDyW), and analysis as follows:
>   - **Results Sum.**:
>     - In comparison, as we mentioned in Line 65-67, our method still shows great accuracy and robustness to aggregation ratio, that our accuracy surpass the baselines at least 2.7/0.5/0.8 on three hard tasks and the avg. decrease is only 0.37 comparing to others about 0.6-1.5.
>     - In the additional experiments, FedPAC[a], the FA of which is the main module pull representations to the global feature centroid, is effective, but is not specifically designed for the overlooked client-sampling information, so it performs relatively not well on some scenarios, e.g., low-aggregation-rate settings, regression tasks, text tasks.
>   - **Analysis**:
>   	- Our method with personalized prior does not have to re-obtain all local information from scratch solely from the data during training. These are especially important on hard-to-learn representations and datasets.
>     - The FA [a] might be affected by the aggregation noise from low ratio, due to the noise makes the global feature centroid [a], unstable and failed.
>     - For example, our method design is crucial for Sent140 from LEAF [11]. The task on Sent140 is a regression task (or binary cls.) on text dataset with heterogeneous input data obtained from social software.  The landscape of rep. on text task is more rugged [53, 16, 13], and the heterogeneous inputs makes it harder.
> 4. [W4] This paper focuses on the specific label distribution problem. How about others?
> - **Clarification**:
>   - Theoretically:
> Our modeling about data $(x_i,y_i)$ in Line 113-119, *the pairs of input and label respectively in dataset $d_i$ on the $i^{th}$ client*, does not depend on a specific labeling distribution. We do not constraint the label in the methodology, framework and convergence analysis.
>   - Experiments:
> In order to verify the effectiveness of pFedBreD on different distributions, we conduct the following experiments as shown in the following table. There are 4 data partition methods, shown in Appendix C.5, and the datasets of our experiments are not on specific label distribution, as follows:
> |Datasets|FEMNIST|Sent140|FMNIST/MNIST/CIFAR-10|$\alpha$-FMNIST|
> |-|-|-|-|-|
> |Partition Methods|Uniform and dominant class [a, 11]|Heterogeneous input data [67, 11]|Rotation allocation|$\alpha$-partition [28] in Table 4|
> |Our performance (Acc.)|70.3|73.7|99.0/93.0/80.6|Avg. 34.8|
> |The best performances among our baselines|66.8|71.3| 98.7/92.2/79.7|Avg. 34.4|
>
> ***Minors***:
> 1. [Q2] The second-best should be 79.68 (Per-FedAvg+FT).
> - Thanks for pointing out, and we will fix it and carefully proof reading of our paper in the revision.
>
> ***References***:
>
> [a] Personalized Federated Learning with Feature Alignment and Classifier Collaboration. ICLR 2023.
>
> [b] Ditto: Fair and Robust Federated Learning Through Personalization. ICML 2021.
>
> [c] Personalized Federated Learning with First Order Model Optimization. ICLR 2021.
>
> [d] Personalized federated learning using hypernetworks. ICML 2021.

---

> > ### Comment · Reviewer_HD18 · 2023-08-16
> >
> > Thanks to the authors for the response. All my concerns have been addressed. I will raise my score.

---

> > > ### Author Response · Authors · 2023-08-17
> > > **Response to the Official Comment**
> > >
> > > We would like to sincerely thank reviewer HD18 for the time and effort he put into this work. We are pleased that our responses address the concerns and have been positively regarded.

---

### Official Review · Reviewer_Q28a · 2023-07-07

**Soundness:** 3 good
**Presentation:** 2 fair
**Contribution:** 2 fair
**Rating:** 6
**Confidence:** 3

**Summary:**

The paper introduces a novel scheme, pFedBreD, which addresses the incomplete information challenge in personalized federated learning by incorporating personalized prior knowledge into the global model of each client. This is achieved by decoupling the personalized prior from the local objective function and applying Bregman divergence regularization. The proposed approach leverages the Expectation-Maximization (EM) algorithm to approximate complete information from both global and local clients. Extensive empirical evaluations on multiple benchmarks demonstrate the effectiveness of the proposed method, and a theoretical analysis is provided to support its theoretical foundations.

**Strengths:**

1. The paper is well-written and easy to read.
2. This paper introduces a novel approach that incorporates Bregman divergence and leverages the Expectation-Maximization (EM) algorithm to approximate complete information from both global and local clients. It effectively enhance personalized federated learning.
3. The effectiveness of the proposed method is validated through extensive empirical evaluations and thorough theoretical analysis in the paper.
4. This work showcases superior performance across multiple benchmarks when compared to various baseline methods.

**Weaknesses:**

1. The authors discuss the concepts of prior knowledge injection and extraction in Figure 1, as well as analyze the process of information injection and extraction. To enhance clarity, it would be helpful if the authors explicitly indicate in the framework design section which step corresponds to these specific actions.
2. Certain aspects of the paper would benefit from additional clarifications to enhance understanding. See below.

**Questions:**

1. Is the method of calculating GCE mentioned in the paper? Does it depend on model parameters or extracted feature maps?
2. Some abbreviation is not clearly explained in the paper. For example, what is MCLR, FT, AM, etc.?
3. How does the computation cost of the proposed method compare to other approaches? Is the local training phase highly time-consuming? For example, what computational resources were utilized in the experiment, and what was the duration required to complete the experiment?
4. Does the scalability of the proposed method remain unaffected? For instance, can the proposed method handle a scenario with more than 100 clients?

**Limitations:**

The authors have adequately addressed the limitations in the paper.

---

> ### Author Rebuttal · Authors · 2023-08-09
>
> ***Comment***:
> - We sincerely thank the reviewer Q28a for the appreciation and constructive comments to further improve clarity and readability, which are detailed in [general responses](https://openreview.net/forum?id=kuxu4lCRr5&noteId=WzbbMONDyW)[W1]. The concern raised by the reviewer, i.e., the further details of our experiments, most of which are in the Appendix, and we answer the reviewer's questions one by one in the hope that our response can address the concern and any questions asked.
>
> ***Responses***:
>
> 1. [W1] Modifications for improving readability.
>  Thanks for the suggestion, and we will make the followings based on the suggestion. Specifically, the motivation and framework illustration, Fig. 1, will be more visually linked to the relevant parts of the math. methodology and computation framework. We will put the algorithm 1 in Appendix into the main body and highlight the relevant parts in the algorithm box with statements.
>
> 2. [Q1, W2] How do we calculate the Generalized Coherence Estimate (GCE)?
> - **What is GCE?** GCE is a classic method [e, 24] in the field of signal processing used to measure the coherence among a batch of sources. This method is mentioned by citing reference [e, 24] in the paper. Formally, $GCE(X) := 1 - \text{det}(\hat{G}(X))$, where $\text{det}(\hat{G}(X))$ is the determinant of the Gram Matrix (Gramian) of $X$. The Gram Matrix consists of the cosine values of each pair of sources in the set $X$. *More details will be included* in the next version of the paper.
> - **How to calculate GCE in this paper**: The GCE in Fig. 2 is $GCE(\{\nabla F_{i}(w_{i})\}_{i})$, as shown in the title of Fig. 2. The GCE analysis is on the first-order directions among each client, which are the gradients of the local objective function. The directions determine the update direction of the global model through aggregation, so our analysis depends on these directions.
>
> 3. [Q3, W2] What about computation cost?
> - Computation cost on both of the local training and entire algorithm *are not highly time-consuming*, see details as follows:
>   - **Entire Alg. Complexity**: the complexity comparison among our baselines in Line 572-578 in Appendix is detailed in Table below, where $N$,$T$,$R$,$K$ and $M$ are respectively the number of clients, global epochs, local epochs, proximal solver iterations and components on each client. Our method *share the same complexity* with pFedMe and FedAMP.
>   - **Local Computation Complexity**: local computation cost is independent of the number of clients $N$. As shown in Table below and the last column of Table 2 in Line 269-271 of the paper, the local computation *only adds constant time* about $N$. Comparing with pFedMe and FedAMP, additional local computation is at most *one extra local gradient and addition computation* for our MAML-based personalized prior strategies. In practice, the local proximal solver iterations *$K$ is not too large*, e.g., $K=5$ in our settings.
> - Main details about Computation Environment for our computation and test:
>   - GPU: TitanX with 1417 Hz frequency, 12288MB mem. 480GB/s of bandwidth and 11 TFLOPs;
>   - System: Ubuntu 20.04.1;
>   - Torch: 1.11.0+cu102.
>
>  [It can be seen that our method improves significantly without great costs. For ease-to-read, we sort by test performance.]
>  |Complexity/Methods|FedEM|FedAvg|pFedMe|Per-FedAvg|FedAMP|pFedBreD (ours)|
>  |-|-|-|-|-|-|-|
>  |Entire Sys. Memory|$O(NM)$|$O(N)$|$O(N)$|$O(N)$|$O(N)$|$O(N)$|
>  |Entire Sys. Time|$O(NTRM)$|$O(NTR)$|$O(NTRK)$|$O(NTR)$|$O(NTRK)$|$O(NTRK)$|
>  |Local Computation Comparing to classic FedAvg|Each Local Loss and Weighted Sum|-|Addition|Gradient|Weighted Sum and L2-Distance|Gradient and Addition|
> |Time Cost of Each Global Epoch with $R=20$ on FMNIST-DNN with TITANX (sec.)|2.7|1.9|3.1|3.3|3.1|3.6|
>  |Average Acc. on All Tasks|71.88|72.80|79.48|80.02|81.80|83.09|
>
> 4. [Q4, W2] Does the scalability of our method remain unaffected, e.g., the case with more than 100 clients?
> - **In our Experiments**: the numbers of clients in the settings of our experiments *are above 100, as shown in Line 228*. The actual time comparison is shown in the Table above in Response 3 [Q3, W2], and the scalability about *algorithm complexity* of the proposed method *is affected linearly* by the number of clients $N$ (linear complexity, an acceptable system expansion cost.).
> - **Theoretically**: according to Theorem 1 in Line 711 in Appendix, we have the linear convergence rate $O(\frac{\hat{S}^{-1}-1}{NT})$, if the client sampling ratio $\hat{S}\le 1$ is fixed, and the convergence rate will *still increase linearly* by $N$.
> - **Conclusion**: the scalability about *both algorithm complexity and convergence* of the proposed method is affected linearly (acceptable linear complexity) by $N$.
>
> ***Minors***:
> 1. [Q2] Some abbreviation is not clearly explained in the paper. For example, what is MCLR, FT, AM, etc.?
> - Thanks for pointing out. They are in the Appendix C1 and C4, and for readability, we will put them in the main body of the revision as suggested.
>
> ***References***:
>
> [e] An Invariance Property of the Generalized Coherence Estimate. TSP 45.4 (1997): 1065-1067.

---

> > ### Comment · Reviewer_Q28a · 2023-08-12
> >
> > Thank you for your detailed response and all my questions are clearly addressed.
> >
> > Having reviewed the general responses and discussions from other reviewers, I will raise my score.

---

> > > ### Author Response · Authors · 2023-08-12
> > > **Response to the Official Comment**
> > >
> > > Thanks for the positive feedback and acknowledgment. Feel free to discuss with us and we will carefully address concerns that may be raised during the discussion phase. In the meantime, we will continue to polish our work based on the feedback from all the reviewers. Once again, thanks for the time and efforts on this work.

---

### Author Rebuttal · Authors · 2023-08-09

# General Responses:
We begin by making the following responses about the structure of our paper and results of additional experiments, and later to allow more space for responses to each author.

- More intuitive and high-level line of logic, which may help understanding and reading, are as follows [(j), the related section j]:
  - **Practical Problem (1): overlooked client-sampling information problem $\rightarrow$ Math. Problem Modeling (4): Bayesian modeling global problem as Maximize Likelihood Estimation (MLE) $\rightarrow$ Optimization Problem Modeling and Framework (4 & 4.1): Introduction of incomplete information and bi-level optimization with Expectation Maximization (EM) $\rightarrow$ Optional Implemental Strategies (4.2): Relaxing mirror descent $\rightarrow$ Computation Problem Formulation and Framework (5.1 & 5.2): Maximum A-Posteriori Estimation (MAP) as local problem and first-order methods $\rightarrow$ Implemental Principle and Computable Algorithms (5.3): Maximum entropy rule and meta step $\rightarrow$ Convergence Analysis (5.4): Bounded errors $\rightarrow$ Numerical Experiments and Analyses (6): 5 analyses for evaluation.**
- Detailed motivation logic is shown as follows:
  1. **Background**: we identify the problem of missing information in prior knowledge in *a single global model* and propose the concept, personalized prior, for this problem in Section 1. Formal discussion, global model $w=E_{i}w_{i}=E_{i}w_{i}|i \Rightarrow$ mutual information (MI) $I(w;i)=0$, is in Line 119-124.
	[From a Bayesian and info. perspective, the global knowledge transferred in conventional method with single global model *has no MI with client-sampling $i$*, in particular when applying reg. $R(w^{(t)};...)$ or local init. $w_{i,0}^{(t)}\leftarrow w^{(t)}$  where $w_{i}$ is the local model on the $i^{th}$ client. This makes the specific model on each client have to re-obtain this information from scratch solely from the data during training, especially impacted on hard-to-learn representations and datasets.]
  2. **Modeling**: we turn this practical problem into a math. one through Bayesian modeling.
  3. **Main Assumption**: *under X-family prior assumption for Bayesian modeling*, Bregman Divergence is introduced, in order to *widen the range of the theory and simplify the computation*.
  4. **Optim. Prob. and Framework**: based on EM, a bi-level optim. prob. and framework. pFedBreD is proposed.
  5. **Implemental Strategies**: a class of strategies called *RMD is proposed as a class of optional implementations* of pFedBreD for personalization.
  6. **Comp. Prob. and Framework**: maximum entropy rule and first-order method for computation is employed to implement the framework as pFedBreD$_{ns}$.
  7. **Experiment Results**: our methods with meta-step strategies reaches the SOTA on 8 public benchmarks and are especially great of our hybrid methods pFedBreD$_{ns, mh}$ *in hard tasks with small aggregation ratios and non-convex local objective settings*.
  8. **Further Impacts**:
     - **Overall**, we introduce the problem of information overlooking during global knowledge transfer in conventional FL.
     - **Theoretically**, our work provides a Bayesian explanation for regularization-based FL methods, modeling and convergence analysis for most of the methods that can be generalized to Bregman regularization.
     - **Practically**, our algorithm gives a new class of the SOTA methods and provides validation from more than 5 different perspectives to demonstrate the effectiveness of PFL in Section 6.2.

- More comparison the reviewer HD18 interested in:
  - We compare the test accuracy between our method and the baselines recommended by reviewer HD18 on hard tasks mentioned in the paper as shown below. The settings are the same as the ones in the paper. We provide an extra column about average decrease for overall comparison about robustness to the noise caused by lowering aggregation ratio. (Note that *lower the aggregation ratio*, *larger the aggregation noise*.)
|Methods/Datasets & Models|FEMNIST & DNN|CIFAR-10 & CNN|Sent140 & LSTM|Average Decrease by Noise|
|:-|-|-|-|:-:|
|**Aggregation Ratio**|**10% $\rightarrow$ 5%**|**20% $\rightarrow$ 10%**|**40% $\rightarrow$ 20%**| - |
|FedPAC[a]|62.2 $\rightarrow$ 60.7|78.9 $\rightarrow$ 77.3|68.1 $\rightarrow$ 66.8|1.5|
|FedHN[d]|61.1 $\rightarrow$ 59.6|77.5 $\rightarrow$ 76.9|71.2 $\rightarrow$ 70.1|1.1|
|Fedfomo[c]|60.1 $\rightarrow$ 58.9|71.4 $\rightarrow$ 70.6 |70.1 $\rightarrow$ 68.9|1.1|
|Ditto[b]|52.9 $\rightarrow$ 52.2|72.4 $\rightarrow$ 72.1|71.0 $\rightarrow$ 70.3| 0.6 |
|mh (ours)|**64.9** $\rightarrow$ **64.3**|**79.4** $\rightarrow$ **79.1**|**72.0** $\rightarrow$ **71.8**| **0.37** |

- Modifications for further improvement of readability from constructive suggestions. More about this are welcomed to discuss in discussion period, and we consider the followings:
  - The pFedBreD alg. in Appendix A.5 will be put between Line 169 and Line 170 in the main body of the paper from the appendix. [[qY3t](forum?id=kuxu4lCRr5&noteId=KNVaTFMM8K)]
  - The Theorems in Appendix D.1 will be put in Section 5 before Remark 1. [[qY3t](forum?id=kuxu4lCRr5&noteId=KNVaTFMM8K)]
  - All the full names of the acronyms will be placed before or around the acronyms. [[qY3t](forum?id=kuxu4lCRr5&noteId=KNVaTFMM8K)]
  - The Fig. 1 will be more visually linked to the various relevant parts of the math. methodology and comp. framework. [[Q28a](forum?id=kuxu4lCRr5&noteId=uuT3KKlAUX)]
  - We will add a subsection to formalize the overlooked information problem. [[jGgs](forum?id=kuxu4lCRr5&noteId=XWZD1FWgxw)]

**References**:

[a] Personalized Federated Learning with Feature Alignment and Classifier Collaboration. ICLR 2023.

[b] Ditto: Fair and robust federated learning through personalization. ICML 2021.

[c] Personalized Federated Learning with First Order Model Optimization. ICLR 2021.

[d] Personalized federated learning using hypernetworks. ICML 2021.

---

### Decision · Program_Chairs · 2023-09-21

**Decision:**

Accept (poster)

**Comment:**

This paper provides a new framework for personalization in federated learning. All reviewers appreciated the clarity of the paper, the novelty of the problem setup/approach, and the extensive experiments that demonstrated the effectiveness of the proposal. There were some concerns about baselines, and presentation issues that have been addressed in the rebuttal by the authors. All reviewers have reached consensus that the paper should be accepted. As such, the paper is recommended to be accepted subject to the authors incorporating the new results from rebuttal into the paper. Congratulations!